# Bioactive Compounds and Potential Health Benefits through Cosmetic Applications of Cherry Stem Extract

**DOI:** 10.3390/ijms25073723

**Published:** 2024-03-27

**Authors:** Abigail García-Villegas, Álvaro Fernández-Ochoa, María Elena Alañón, Alejandro Rojas-García, David Arráez-Román, María de la Luz Cádiz-Gurrea, Antonio Segura-Carretero

**Affiliations:** 1Department of Analytical Chemistry, University of Granada, Av. Fuentenueva s/n, 18071 Granada, Spain; abigarcia@ugr.es (A.G.-V.); alvaroferochoa@ugr.es (Á.F.-O.); alejorogar@ugr.es (A.R.-G.); darraez@ugr.es (D.A.-R.); ansegura@ugr.es (A.S.-C.); 2Regional Institute for Applied Scientific Research (IRICA), University of Castilla-La Mancha, Avda. Camilo José Cela 10, 13071 Ciudad Real, Spain; mariaelena.alanon@uclm.es; 3Department of Analytical Chemistry and Food Science and Technology, University of Castilla-La Mancha, Ronda de Calatrava 7, 13071 Ciudad Real, Spain

**Keywords:** cherry, revalorization, by-products, bioactive compounds, antioxidants, anti-aging, cosmeceuticals

## Abstract

Cherry stems, prized in traditional medicine for their potent antioxidant and anti-inflammatory properties, derive their efficacy from abundant polyphenols and anthocyanins. This makes them an ideal option for addressing skin aging and diseases. This study aimed to assess the antioxidant and anti-inflammatory effects of cherry stem extract for potential skincare use. To this end, the extract was first comprehensively characterized by HPLC-ESI-qTOF-MS. The extract’s total phenolic content (TPC), antioxidant capacity, radical scavenging efficiency, and its ability to inhibit enzymes related to skin aging were determined. A total of 146 compounds were annotated in the cherry stem extract. The extract effectively fought against NO^·^ and HOCl radicals with IC_50_ values of 2.32 and 5.4 mg/L. Additionally, it inhibited HYALase, collagenase, and XOD enzymes with IC_50_ values of 7.39, 111.92, and 10 mg/L, respectively. Based on the promising results that were obtained, the extract was subsequently gently integrated into a cosmetic gel at different concentrations and subjected to further stability evaluations. The accelerated stability was assessed through temperature ramping, heating-cooling cycles, and centrifugation, while the long-term stability was evaluated by storing the formulations under light and dark conditions for three months. The gel formulation enriched with cherry stem extract exhibited good stability and compatibility for topical application. Cherry stem extract may be a valuable ingredient for creating beneficial skincare cosmeceuticals.

## 1. Introduction

Aging is an irreversible process that occurs with declining cell and tissue functions and an increased risk of age-related diseases, including neurodegenerative diseases, metabolic diseases, cardiovascular diseases, and skin diseases, among others [1]. Research is currently focused on finding treatments that can reduce the development of age-related diseases and, in turn, promote healthy aging [2].

One of the main concerns of today’s society is skin aging. Skin aging is an inevitable biological process of human life and is a consequence of advancing age [3]. The signs of aging begin to appear with changes in the structure, function, and appearance of the skin. These changes result in skin that is drier, thinner, and at greater risk of skin disorders. Skin aging is caused by both intrinsic and extrinsic factors [4]. The main causes of intrinsic skin aging are reactive oxygen species (ROSs) generated continuously as natural by-products of mitochondrial metabolism, the decrease in the replicative capacity of keratinocytes, fibroblasts, and melanocytes over time, and the increase in the expression of enzymes that degrade the extracellular matrix (ECM) of the dermis [5]. However, the skin is not only affected by the intrinsic aging of the organism, but also by extrinsic factors that promote skin aging [3]. Among these extrinsic factors, ultraviolet (UV) radiation is the main cause of skin aging and is responsible for the generation of ROSs that cause damage to DNA, lipids, and proteins [4,5]. UVA radiation penetrates deep into the skin, damaging connective tissue and increasing the risk of skin cancer, while also contributing to skin aging [6]. UVB radiation affects the epidermis, causing sunburn and leading to photoaged skin with increased fragility, scarring, and hyperpigmentation [6].

In this context, the adult population are increasingly aware of their health and demand cosmetic products and formulations to improve their skin condition [7]. This demand increases the need to develop new skin care products containing active ingredients that help to combat the signs of aging [8]. Therefore, the cosmetic industry is increasingly researching various active ingredients for the development of new natural, safe, and effective products that provide benefits to the skin [7].

As a result of research in this sector, it has been shown that products of natural origin, such as waste from the agri-food industry, contain compounds that could be revalorized for cosmetic applications. In addition, the demand for natural products together with the desire to respect the environment suggests the need to exploit these wastes and by-products to obtain beneficial ingredients for skin health [7,9].

Many agro-industrial by-products are rich in phenolic compounds [10]. Polyphenols have been shown to have antioxidant and anti-inflammatory properties and may be ideal for preventing and mitigating various skin disorders [10,11]. In the skin, phenolic compounds act through various mechanisms of action including, for example, the elimination of radical species, prevention of cellular senescence, inhibition of dermal proteinases, and inhibition of inflammatory processes [5]. In addition, these compounds have been found to stimulate collagen production in fibroblasts and are involved in the regulation of wound healing [12]. In vivo studies have demonstrated the effects of polyphenol-rich botanical extracts on the skin [13]. For example, grape stem and seed extracts, rich in polyphenols such as catechin, resveratrol, and gallic acid, have been shown to be excellent antioxidant and anti-aging agents [14,15]. It has also been observed that green tea extract and mango peel extract, rich in epigallocate-chin-3-gallate, are able to inhibit inflammatory enzymes and cytokines, and that quercetin-rich extracts such as passion fruit peel and seeds block UV-induced inflammation [16,17,18].

In this article we focus on the cherry stem because of its excellent antioxidant properties. The cherry of the genus *Prunus avium* is a fruit widely traded in the world, characterized by its nutritional value and its high content of phytochemicals with antioxidant and anti-inflammatory properties [19]. Currently, cherries are widely cultivated in the Mediterranean area due to its temperate climate [20]. In addition to fresh consumption, cherries are used by the agri-food industry, usually to produce juices and jams [20]. As a result of processing, agri-food industries generate large quantities of waste, mainly leaves, bones, and stems [21]. These by-products, which are usually discarded, are an important source of phytochemicals [20]. In ancient times, cherry stems were used as an analgesic in the form of an infusion for many years in folk medicine [22]. The cherry stem has a high content of health-promoting bioactive compounds, mainly phenolic compounds such as flavonoids, phenolic acids, and anthocyanins [23,24,25]. They are also rich in saponins, which are responsible for the bitter taste and hypocholesterolemic effects of cherry stems [20]. Therefore, the re-evaluation of this by-product could be of interest for obtaining bioactive ingredients for subsequent incorporation in cosmetics and obtaining high value-added products.

In this context, this study aimed to evaluate cherry stems as a potential source of phytochemicals and to explore their revalorization to obtain bioactive compounds with skin benefits as well as their possible use in cosmeceutical formulations. To achieve this objective, the phytochemical composition of the extract was characterized, a comprehensive in vitro investigation was carried out to measure its total phenolic content (TPC), its antioxidant capacity was evaluated, and its ability to inhibit ROSs/RNSs and enzymes was determined. Additionally, a cosmetic prototype containing cherry stem extract was formulated and subjected to stability assessments. At present, few studies have been conducted on the benefits of cherry stem extract on the skin. This is the first time that cherry stem extract has been incorporated into a cosmetic formulation as a bioactive ingredient and its stability in the formulation has been studied. Therefore, this study aimed to investigate the use of cherry stem extract as a novel ingredient in the world of cosmeceuticals through the preliminary development of a stable cosmetic product with potential skin benefits.

## 2. Results and Discussion

### 2.1. Phytochemical Characterization

The HPLC-ESI-qTOF-MS analysis allowed the detection of 174 molecular features after the data processing steps. Among these signals, 146 could be provisionally annotated, and the remaining 28 remained as unknowns. Figure 1 shows the chromatogram of the base peak. The cherry stem extract was shown to be an extract rich in compounds and a high number of signals were annotated. More compounds were annotated in this work than in previously reported articles from this matrix [23,24,26,27]. Table 1 lists the annotated compounds organized by their respective retention times (RT).

Among the detected compounds, phytochemicals belonging to the families of organic acids, cinnamic acids, hydroxybenzoic acids, flavonoids, terpenoids, phenylpropanoids, aliphatic compounds, carbohydrates, and fatty acids were detected. Considering their relative abundances, flavonoids were the most relevant group in the cherry stem extract, mainly the subclass flavan-3-ols, flavones, and proanthocyanidins. The presence of (epi)catechin and type A, B, and C procyanidins was tentatively detected. Proanthocyanidins are phenolic compounds found mainly in plant stems and seeds that have been characterized for their antioxidant and anti-inflammatory properties [28,29]. Epicatechin and proanthocyanidins have been previously described in cherry stems [27]. Naringerin, aromadendrin, apigenin, hesperetin, quercetin, isoquercetin, isorhamnetin, and their glycosides were also tentatively annotated. Naringerin and aromadendrin were observed abundantly in the extract. Among the cinnamic acids, glycosylated hydroxycinnamic acids, such as chlorogenic acid and ciic acid, stood out. Previously, other studies had reported the presence of chlorogenic acid and coumaric acid hexosides in cherry stem extract [30]. Furthermore, the high-resolution mass spectrometry method also allowed for the tentative characterization of hydroxybenzoic acid derivatives, such as protocatechuic acid glycosides, vanillic acid, and vanillic acid glycosides, whose presence in the extract was abundant. Therefore, the chromatogram of the cherry stem extract showed abundant signals. The most abundant signals in the chromatogram were protocatechuic acid, (epi)catechin, betulalbusida A, naringenin, chrysoeriol glucoside, rhodioloside B, dimethoxyapigenin, and methylnaringenin. The compounds betulalbuside A and rhodioloside B were annotated for the first time in cherry stem extract. A total of 62 compounds were annotated for the first time in cherry stem extract, and some of them were: junipediol A glucoside, roseoside, icariside D1, linalool oxide primeveroside, segetoside A, cinnamoyl arabinosylglucose, vitexin, multifidol apiosyl glucoside, dihydrophaseic acid glucopyranoside, trihydroxydihydrochalcone glucoside, cudranian 2, coumaroyltrifolin, gingerglycolipid A, liquiritic acid, ginsenoside Rh2, and soyacerebroside I.
ijms-25-03723-t001_Table 1Table 1Annotated phytochemical compounds in cherry stem extract using HPLC-ESI-qTOF-MS, including retention times (RT), experimental and theoretical *m/z* values, ppm errors, annotation level, molecular formula, MS/MS fragments, relative abundance, and references.PeakRT(min)*m*/*z*Experimental[M-H]^−^*m*/*z*Theoretical[M-H]^−^Error(ppm)Level ofAnnotationMolecular FormulaProposed CompoundMS/MS FragmentsRel. Ab.(%)References10.91131.0459131.0462−2.692C_4_H_8_N_2_O_3_L-Asparagine1310.45PubChem: 626720.96181.0717181.0717−0.552C_6_H_14_O_6_D-Manitol59, 71, 85, 89, 101, 110, 1811.63PubChem: 578031.01195.0514195.05101.732C_6_H_12_O_7_D-gluconic acid75, 99, 129, 1953.54[26]41.06133.0146133.01422.382C_4_H_6_O_5_Malic acid isomer 171, 115, 1330.91[31]51.11133.0147133.01423.132C_4_H_6_O_5_Malic acid isomer 271, 115, 1330.91[31]61.16133.0147133.01423.132C_4_H_6_O_5_Malic acid isomer 371, 115, 1331.91[31]71.20191.0215191.01979.092C_6_H_8_O_7_Citric acid57, 85, 87, 110, 1911.91[31]81.30197.8079--2-Unknown-0.82-91.39125.0000125.0009−7.652C_2_H_7_O_4_PDimethylphosphate62, 780.76HMDB0061734101.60115.0034115.0037−2.772C_4_H_4_O_4_Maleic acid990.71HMDB0000176111.9496.960896.96016.812H_2_O_4_SSulfate79, 960.90HMDB0001448122.13358.1142358.1143−0.542C_15_H_21_NO_9_L-DOPA glucoside101, 150, 161, 3120.33HMDB0029452132.29299.0765299.0772−2.602C_13_H_16_O_8_Hydroxybenzoic acid glucoside93, 116, 2990.15[23]142.58315.0730315.07212.562C_13_H_16_O_9_Protocatechuic acid glucoside isomer 1153, 109, 271, 3150.84[26]152.87315.0729315.07212.242C_13_H_16_O_9_Protocatechuic acid glucoside isomer 2153, 109, 271, 3152.32[26]163.04313.0934313.09291.512C_14_H_18_O_8_Mandelic acid glucopyranoside isomer 159, 71, 73, 85, 101, 161, 269, 3130.46HMDB0038335173.39313.0933313.09291.192C_14_H_18_O_8_Mandelic acid glucopyranoside isomer 259, 71, 73, 85, 101, 161, 269, 3131.17HMDB0038335183.55353.0878353.0878−0.122C_16_H_18_O_9_Chlorogenic acid isomer 185, 87, 93, 134, 135, 179, 191, 3530.37[23]194.10353.0881353.0878−0.122C_16_H_18_O_9_Chlorogenic acid isomer 285, 87, 93, 134, 135, 179, 191, 3530.57[23]204.35341.0884341.08781.632C_15_H_18_O_9_Caffeic acid glucoside133, 161, 179, 281, 3410.22[23,31]214.47359.1349359.13470.302C_16_H_24_O_9_Junipediol A glucoside131, 293, 313, 3590.41CNP0148585224.62269.1015269.1030−5.942C_13_H_18_O_6_Benzyl glucopyranoside71, 85, 101, 161, 2690.05CNP0261916235.01329.0871329.0878−2.262C_14_H_18_O_9_Vanillic acid glucoside isomer 196, 129, 1730.56[27]245.27329.0870329.0878−2.562C_14_H_18_O_9_Vanillic acid glucoside isomer 296, 129, 1730.56[27]255.47401.1442401.1453−2.892C_18_H_26_O_10_Benzyl primeveroside isomer 1101, 161, 269, 4010.16[26]265.59289.0712289.0717−2.072C_15_H_14_O_6_(Epi)Catechin isomer 197, 109, 123, 137, 203, 205, 245, 289, 2051.63[23,31]275.93401.1442401.1453−2.892C_18_H_26_O_10_Benzyl primeveroside isomer 2101, 161, 269, 4010.48[26]286.12289.0711289.0717−2.422C_15_H_14_O_6_(Epi)Catechin isomer 297, 109, 123, 137, 203, 205, 245, 2893.27[23,31]296.27325.0907325.0929−6.852C_15_H_18_O_8_Coumaroyl glucosa isomer 1117, 119, 145, 163, 289, 3250.46[23]306.48577.1328577.1351−4.142C_30_H_26_O_12_Procyanidin B isomer 1125, 245, 289, 407, 425, 577, 5780.79[26]316.58385.1864385.1868−1.112C_19_H_30_O_8_Roseoside isomer 159, 71, 89, 101, 3850.33CNP0241893326.71385.1864385.1868−1.112C_19_H_30_O_8_Roseoside isomer 259, 71, 89, 101, 3850.72CNP0241893337.00385.1864385.1868−1.112C_19_H_30_O_8_Roseoside isomer 359, 71, 89, 101, 3851.91CNP0241893347.14577.1336577.1351−2.752C_30_H_26_O_12_Procyanidin B isomer 2125, 245, 289, 407, 425, 577, 5780.48[26]357.42449.1091449.10890.292C_21_H_22_O_11_Aromadendrin glucoside isomer 1125, 151, 287, 4491.25[23,31]367.61325.0921325.0929−2.552C_15_H_18_O_8_Coumaroyl glucose isomer 2117, 119, 145, 205, 325, 1631.36[23]377.73289.0715289.0717−1.032C_15_H_14_O_6_(Epi)catechin isomer 397, 109, 123, 137, 203, 205, 245, 289, 2050.87[31]387.81449.1099449.10892.072C_21_H_22_O_11_Aromadendrin glucoside isomer 2125, 151, 287, 4490.23[31]398.00415.1590415.1610−4.842C_19_H_28_O_10_Icariside D159, 71, 89, 101, 131, 149, 191, 4150.14HMDB0303758408.07577.1344577.1351−1.362C_30_H_26_O_12_Procyanidin B isomer 3125, 245, 289, 407, 425, 577, 5780.83[26]418.15391.1608----Unknown-0.74-428.22865.1977865.1985−1.012C_45_H_38_O_18_Procyanidin C isomer 1125, 161, 243, 289, 287, 407, 577, 695, 713, 865, 8660.71HMDB0038370438.27463.2178463.2185−1.562C_21_H_36_O_11_Linalool oxide primeveroside isomer 159, 89, 131, 149, 4630.89HMDB0035489448.41331.1758331.1762−1.402C_16_H_28_O_7_Betulalbusida A59, 71, 89, 119, 179, 3310.02HMDB0035634458.49463.2178463.2185−1.562C_21_H_36_O_11_Linalool oxide primeveroside isomer 259, 89, 131, 149, 4630.03HMDB0035489468.54865.1972865.1985−1.592C_45_H_38_O_18_Procyanidin C isomer 2125, 161, 243, 289, 287, 407, 577, 695, 713, 865, 8660.16HMDB0038370478.62463.2178463.2185−1.562C_21_H_36_O_11_Linalool oxide primeveroside isomer 359, 89, 131, 149, 4630.31HMDB0035489488.67519.1729519.17191.802C_22_H_32_O_14_Segetoside A147, 161, 325, 4730.22CNP0159091498.76441.1408441.14021.202C_20_H_26_O_11_Cinnamoyl arabinosyl glucose103, 147, 161, 4410.53HMDB0030294508.81865.1988865.19850.262C_45_H_38_O_18_Procyanidin C isomer 3125, 161, 243, 289, 287, 407, 577, 695, 713, 865, 8660.46HMDB0038370518.88771.1995771.19890.692C_33_H_40_O_21_Quercetin rutinoside glucoside151, 179, 300, 301, 609, 7710.37[23,31]528.92755.2018755.2040−2.982C_33_H_40_O_20_Kaempferol rutinoside glucoside125, 285, 465, 466, 593, 594, 7550.16[23,31]538.99465.1050465.10382.392C_21_H_22_O_12_Taxifolin glucoside125, 152, 259, 285, 437, 4650.94[23,31]549.09521.2026521.2028−0.522C_26_H_34_O_11_Icariside E5 isomer 1329, 341, 359, 521, 5670.16HMDB0034749559.14521.2026521.2028−0.522C_26_H_34_O_11_Icariside E5 isomer 2329, 341, 359, 521, 5670.25HMDB0034749569.23431.0990431.09841.382C_21_H_20_O_10_Vitexin117, 283, 311, 341, 4310.37CNP0212195579.27473.1682473.16643.622C_21_H_30_O_12_Feruloylglucose trihydroxy methylbutylglycoside89, 125, 149, 293, 427, 4730.16HMDB0036214589.38463.1243463.1246−0.702C_22_H_24_O_11_Hesperetin glucoside109, 165, 257, 301, 343, 435, 4630.42[32]599.46503.1761503.1770−1.892C_22_H_32_O_13_Multifidol apiosyl glucoside isomer 159, 71, 89, 125, 148, 149, 131, 293, 457, 5030.52HMDB0039930609.52595.1658595.1668−1.702C_27_H_32_O_15_Naringenin diglucoside271, 433, 5950.01[26]619.58609.1475609.14612.222C_27_H_30_O_16_Rutin255, 271, 300, 301, 609, 610, 6110.74[26]629.65503.1776503.17701.092C_22_H_32_O_13_Multifidol apiosyl glucoside isomer 259, 71, 89, 125, 148, 149, 131, 293, 457, 5030.27HMDB0039930639.75303.0519303.05102.762C_15_H_12_O_7_Taxifolin125, 153, 177, 259, 275, 285, 3030.50[32]649.82463.0886463.08820.782C_21_H_20_O_12_Isoquercetin271, 300, 3010.42[31]659.90433.1153433.11402.972C_21_H_22_O_10_Naringenin glucoside isomer 183, 107, 119, 151, 165, 253, 271, 4331.50[23,31]669.92477.1051477.10382.542C_22_H_22_O_12_Isorhamnetin glucoside119, 151, 269, 271, 299, 4310.67[26]679.99431.0995431.09842.542C_21_H_20_O_10_Cosmosiin isomer 1125, 133, 269, 4310.05HMDB00373406810.01593.1520593.15121.302C_27_H_30_O_15_Kaempferol rutinoside isomer 1593, 285, 255, 227, 2840.67[31]6910.05433.1156433.11403.672C_21_H_22_O_10_Naringenin glucoside isomer 283, 107, 119, 151, 165, 253, 271, 4330.41[31]7010.11307.0464307.04591.374C_14_H_12_O_8_Fulvic acid69, 79, 99, 141, 185, 3070.30CNP02497687110.17443.1902443.1923−4.762C_21_H_32_O_10_Dihydrophaseic acid glucopyranoside59, 71, 133, 311, 443, 445, 4890.14CNP02101177210.20417.1197417.11911.342C_21_H_22_O_9_Liquiritin211, 237, 4170.35[26]7310.25431.0973431.09843.232C_21_H_20_O_10_Cosmosiin isomer 283, 125, 149, 253, 268, 4310.15HMDB00373407410.29447.1077--4-Unknown-0.45-7510.40433.1147433.11401.592C_21_H_22_O_10_Naringenin glucoside isomer 383, 107, 119, 151, 165, 253, 271, 4330.75[31]7610.46329.1609329.16060.872C_16_H_26_O_7_Carboxyethyl dihydroxycyclopentyl oxooctanoic acid59, 225, 269, 3290.15CNP01509117710.48433.1152433.11402.742C_21_H_22_O_10_Naringenin glucoside isomer 457, 83, 125, 149, 253, 271, 4330.13[31]7810.51463.2527--4-Unknown-0.19-7910.57575.1200575.11950.812C_30_H_24_O_12_Procyanidin A125, 137, 271, 394, 449, 5750.29HMDB00376558010.61595.1666595.1668−0.472C_27_H_32_O_15_Neoeriocitrin255, 256, 549, 5500.31[32]8110.65493.2290493.2290−0.182C_22_H_38_O_12_Rhodioloside B isomer 1101, 131, 161, 315, 447, 4930.29CNP03289868210.69493.2325493.22906.922C_22_H_38_O_12_Rhodioloside B isomer 2101, 131, 161, 315, 447, 4930.15CNP03289868310.76287.0574287.05614.362C_15_H_12_O_6_Aromadendrin125, 243, 259, 260, 2870.20HMDB00308478410.81835.2469--4-Unknown-1.10-8510.92831.2151--4-Unknown-1.50-8610.96371.1704--4-Unknown-0.29-8711.09461.1097461.10891.582C_22_H_22_O_11_Chrysoeriol glucoside63, 65, 2551.50[32]8811.22447.1293447.1297−0.912C_22_H_24_O_10_Dihydrowogonin glucoside65, 136, 171, 241, 269, 285, 4473.68[31]8911.26491.1209491.11952.782C_23_H_24_O_12_Dimethoxyapigenin glucoside268, 269, 270, 283, 285, 432, 4471.06HMDB00388259011.38419.1335419.1347−3.082C_21_H_24_O_9_Trihydroxydihydrochalcone glucoside isomer 1213, 239, 257, 4190.18HMDB00374839111.42609.1248609.1250−0.352C_30_H_26_O_14_Caffeoylastragalin161, 179, 227, 255, 285, 3230.12HMDB00302399211.61569.1293569.1301−1.412C_28_H_26_O_13_Cudranian 2125, 259, 285, 541, 5690.12CNP00851969311.88301.0721301.07171.002C_16_H_14_O_6_Trihydroxy methoxyflavanone110, 137, 165, 194, 258, 286, 3010.09CNP01882139411.99593.1315593.13012.362C_30_H_26_O_13_Coumaroyltrifolin145, 227, 255, 285, 5930.18HMDB00406899512.03327.2180327.21770.812C_18_H_32_O_5_Trihydroxyoctadecadienoic acid171, 327, 2111.29[27]9612.24419.1354419.13471.452C_21_H_24_O_9_Trihydroxydihydrochalcone glucoside isomer 291, 171, 175, 4190.15HMDB00374839712.49271.0616271.06121.352C_15_H_12_O_5_Naringenin63, 65, 107, 119, 151, 177, 2710.46[27]9812.51329.2332329.2333−0.562C_18_H_34_O_5_Trihydroxyoctadecenoic acid211, 229, 3291.32[27]9913.03269.0461269.04551.922C_15_H_10_O_5_Genistein62, 65, 133, 159, 2690.31HMDB000321710013.15563.1572563.15592.272C_30_H_28_O_11_Dihydroxyphenyl oxopropenylphenoxy trihydroxyoxanylmethyl hydroxyphenyl propenoate145, 211, 307, 5630.13CNP032592810113.34521.1458521.14530.852C_28_H_26_O_10_Morusalbanol A169, 211, 237, 5210.13PubChem: 6016629610213.48517.3163517.3171−1.572C_30_H_46_O_7_Jaligonic acid469, 470, 487, 488, 5170.40[27]10313.59519.3362--4-Unknown-0.22-10413.63307.1917307.19150.592C_18_H_28_O_4_Corchorifatty acid D65, 121, 185, 235, 289, 3070.22HMDB003324310513.76285.0773285.07681.462C_16_H_14_O_5_Methylnaringenin65, 82, 110, 137, 165, 270, 2852.59[27]10613.92255.0665255.06630.712C_15_H_12_O_4_Pinocembrin isomer 165, 169, 211, 2550.18[26]10714.07283.0615283.06120.942C_16_H_12_O_5_narinnin109, 163, 184, 268, 2830.15CNP033079310814.14597.1626597.16142.012C_30_H_30_O_13_Myrciacitrin V241, 269, 286, 327, 536, 5510.09CNP024447310914.19255.0664255.06630.322C_15_H_12_O_4_Pinocembrin isomer 265, 107, 145, 151, 169, 2110.40[26]11014.56253.0510253.05061.312C_15_H_10_O_4_Chrysin63, 65, 143, 145, 2530.75[26]11115.07675.3618675.35973.012C_33_H_56_O_14_Gingerglycolipid A397, 415, 675, 676, 7210.01[17]11215.49513.2867--4-Unknown-0.23-11315.83293.2114293.2122−2.922C_18_H_30_O_3_Hydroxyoctadecatrienoic acid183, 275, 2930.38[26]11415.85487.3417--4-Unknown-0.37-11515.99537.3289--4-Unknown-0.12-11616.15441.2857441.2858−0.262C_24_H_42_O_7_Ascorbyl stearate163, 181, 277, 4410.03HMDB003824211716.22291.1967291.19660.332C_18_H_28_O_3_Oxooctadecatrienoic acid121, 185, 2910.40CNP025696111816.30494.3206494.3252−9.412C_24_H_50_NO_7_PPalmitoyl phosphatidylcholine78, 224, 255, 4800.22HMDB001038211916.53295.2279295.2279−0.022C_18_H_32_O_3_Hydroxy-octadecadienoic acid2950.61[26]12016.58417.2865--4-Unknown-0.15-12116.88293.2109293.2122−4.622C_18_H_30_O_3_Oxooctadecadienoic acid2930.76CNP019156312217.10469.3316469.3323−1.642C_30_H_46_O_4_Liquiritic acid393, 407, 423, 4690.29HMDB003578812317.29565.3598--4-Unknown-0.50-12417.45633.3801633.37970.612C_39_H_54_O_7_Hydroxypyracrenic acid117, 145, 6331.50HMDB002978012517.53471.3488471.34801.652C_30_H_48_O_4_Hederagenin isomer 14710.84CNP017096612617.70471.3502471.34804.622C_30_H_48_O_4_Hederagenin isomer 24710.26CNP017096612717.79621.3495621.34920.483C_30_H_54_O_13_Fatty acid derivate-0.30-12817.87445.3170445.3171−0.262C_24_H_46_O_7_Fatty acid derivate-0.74-12918.10607.3702607.36990.432C_30_H_56_O_12_Stearoyl trehalose59, 103, 163, 6070.65PubChem: 6894554713018.60277.2174277.21730.212C_18_H_30_O_2_Linolenic acid2771.01[26]13118.74663.3616--4-Unknown-0.49-13218.84499.3790499.3793−0.642C_32_H_52_O_4_Oleananoic acid acetate423, 441, 451, 453, 4990.13PubChem: 670866813318.90649.3829649.374612.742C_39_H_54_O_8_Coumaroyloxytormentic acid isomer 159, 101, 145, 205, 251, 6490.35HMDB004068213418.95649.3828649.374612.582C_39_H_54_O_8_Coumaroyloxytormentic acid isomer 259, 101, 145, 205, 251, 6490.25HMDB004068213519.06617.3851617.38480.492C_39_H_54_O_6_Coumaroylalphitolic acid isomer 1117, 145, 6170.23CNP028860413619.16617.3867617.38483.082C_39_H_54_O_6_Coumaroylalphitolic acid isomer 2117, 145, 6170.37CNP028860413719.21617.3868617.38483.242C_39_H_54_O_6_Coumaroylalphitolic acid isomer 3117, 145, 6170.27CNP028860413819.26299.2596299.25921.322C_18_H_36_O_3_Hydroxy octadecanoic acid59, 2990.53[26]13919.30279.2334279.23291.462C_18_H_32_O_2_Linoleic acid2790.54[27]14019.69691.3918691.39101.062C_34_H_60_O_14_Acetyl octadecanoyl trehalose59, 145, 205, 251, 6910.49CNP027801214119.89555.2784555.2811−4.602C_28_H_44_O_11_Picrasinoside F100, 5550.90PubChem: 1473374014220.72935.5802935.57376.862C_51_H_84_O_15_Digalactosyldiacylglycerol277, 379, 397, 657, 675, 9810.01CNP013061214321.12621.4392621.43723.172C_36_H_62_O_8_Ginsenoside Rh2575, 6152.04HMDB003954414421.31429.3026429.30103.562C_27_H_42_O_4_Plastoquinone 1 isomer 1133, 161, 179, 4290.20CNP015265114521.41712.5358712.5369−1.612C_40_H_75_NO_9_Soyacerebroside I isomer 1532, 550, 7120.09CNP024539314621.46665.4191--4-Unknown-0.27-14721.53712.5358712.5369−1.612C_40_H_75_NO_9_Soyacerebroside I isomer 2532, 550, 7120.07CNP024539314821.58429.3023429.30102.872C_27_H_42_O_4_Plastoquinone 1 isomer 2133, 161, 179, 4291.16CNP015265114921.76773.5240773.52093.932C_45_H_74_O_10_Octadecatrienoyl galactosyl glycerol235, 253, 277, 513, 7730.65CHEBI:3404215022.00959.5959--4-Unknown-0.19-15122.18443.3187443.31674.473C_28_H_44_O_4_Hydroxymethoxyphenyl propenyl octadecenoate77, 105, 133, 343, 428, 4430.16CNP027114315222.29863.6428--4-Unknown-0.65-15322.41775.5367775.53660.122C_45_H_76_O_10_Octadeca dienoyloxytrihydroxy(hydroxymethyl)oxanyloxypropyloctadecatrienoate253, 277, 279, 5130.11CNP011082215422.48413.3078413.30613.982C_27_H_42_O_3_Hydroxyphenyl propenyl octadecenoate117, 118, 119, 145, 4130.49CNP023003215522.54461.3293461.32724.373C_28_H_46_O_5_Polyporusterone F57, 125, 4610.13HMDB003849915622.64863.6425--4-Unknown-2.32-15722.87531.4077--4-Unknown-0.18-15823.08531.3739--4-Unknown-0.11-15923.16445.3353--4-Unknown-0.54-16023.35919.7022--4-Unknown-0.11-16123.53415.3224415.32181.432C_27_H_44_O_3_Octadecyl coumarate117, 118, 119, 145, 163, 4150.63CNP015604416223.63489.3605489.35853.912C_30_H_50_O_5_Escinidin57, 83, 125, 4890.61[17]16323.75459.3493459.34802.782C_29_H_48_O_4_Eicosanyl caffeate133, 161, 179, 4590.40PubChem: 532023816424.03659.3641--4-Unknown-0.22-16524.21545.2950545.29097.522C_34_H_42_O_6_Cowagarcinone A176, 369, 527, 5450.11C0003527716624.58824.4111--4-Unknown-0.07-16725.10905.6708--4-Unknown-0.27-16825.14756.5116--4-Unknown-0.07-16925.54887.6596--4-Unknown-0.20-17025.72715.5865715.5882−2.453C_45_H_80_O_6_Triacylglycerol61, 7150.40HMDB004788517125.94759.6139--4-Unknown-0.13-17226.21699.5927699.5933−0.912C_45_H_80_O_5_Diacylglycerol61, 6990.11HMDB000766017326.33729.6028--4-Unknown-0.59-17426.97743.6174743.6195−2.893C_47_H_84_O_6_Tetradecenoyl hexadecenoyl tetradecenoyl glycerol61, 7430.29HMDB0047885RT: retention time.

In addition, the presence of organic acids such as malic acid, maleic acid, and citric acid was also revealed, with malic acid being the most abundant organic acid. Maleic acid was annotated for the first time in cherry stem extract. Previous work on cherry stems showed a smaller number of compounds. For example, Afonso et al. reported only 16 compounds in cherry stem ethanolic extract, Babota et al. annotated 27 compounds for cherry stem extract, and Nastić et al. and Agulló-Chazarra et al. reported 42 and 57 compounds, respectively [26,27,30,33].

### 2.2. Evaluation of Antioxidant and Antiradical Activities

The evaluation of the antioxidant activity of cherry stem extract was performed using different in vitro methods. First, the TPC of the cherry stem extract was quantified using the Folin–Ciocalteu method. This method is widely used to semi-quantify phenolic compounds in plant extracts [34]. The antioxidant activity of the cherry stem extract was then evaluated using the FRAP, TEAC, and ORAC methods. Table 2 shows the TPC value and antioxidant values of the cherry stem extract as obtained by the FRAP, TEAC, and ORAC methods.

The cherry stem extract showed high FRAP, TEAC, and ORAC values, revealing the antioxidant capacity of the extract. The antioxidant values of the cherry stem can be mainly attributed to the presence of flavonoids, since the high presence of this type of compound has been demonstrated in the phytochemical characterization carried out by LC-MS. Flavonoids have demonstrated a potent antioxidant capacity, which allows them to act as hydrogen donors, reducing agents, free radical scavengers, metal ion chelators, and oxygen inhibitors [35]. These antioxidant properties are due to the presence and position of the active ‘‘-OH’’ groups present in their structure. For example, the antioxidant power of procyanidins will depend on their degree of polymerization, so the chelating and free radical scavenging properties of cherry stem extract may be due to the presence of highly polymerized procyanidins [36]. The FRAP value could be related to the presence of organic acids, especially to the presence of chlorogenic acid, which was also annotated in the extract. In addition, the cherry stem extract exhibited a high ORAC value, which could be justified by the presence of protocatechuic acid, catechin, and quercetin in the extract. Protocatechuic acid, catechin, and quercetin all have a catechol group in their structure, which enhances activity against peroxyl radicals [37].

The values obtained have been compared with previous research; however, differences in extraction systems, the nature of samples, and testing methodologies must be considered for good interpretation. In this regard, Bursal et al. previously studied cherry stem extract and obtained a TPC value of 146.5 ± 13.10 mg GAE/g sample [38]. Ademović et al. and Babota et al. reported TPC values of 121.3 ± 9.7 and 37.63 ± 2.75 mg GAE/g sample, respectively for cherry stem extract [33,39]. These authors reported TPC values similar to that obtained in the present study (131.4 ± 0.4 mg GAE/g sample). It should be noted that the cherry corner extract used in the present study was a very rich extract with which 174 signals were annotated and, therefore, may be clearly related to a better TPC value. On the other hand, Demir et al. evaluated the antioxidant capacity of cherry stem extract and obtained FRAP and TEAC values of 0.004 ± 0 mmol FeSO_4_/g sample and 3.02 ± 0.01 µmol TE/g sample, respectively [24]. In addition, the pulp of four cherry cultivars showed TPC values between 0.87 and 1.73 mg GAE/g sample [32]. These values were lower than those obtained for the cherry stem extract.

For a better understanding of the antioxidant capacity of the cherry stem extract, the anti-radical activity was also determined using different reactive oxygen and nitrogen species (ROSs and RNSs).

ROSs and RNSs are produced at the cellular level during metabolic processes, but can also be induced extrinsically by UV rays, pollutants, and stress [40]. Physiologically, the human body requires low levels of ROSs and RNSs for proper functioning; however, very high levels of ROSs and RNSs can lead to a state of oxidative stress and accelerate the aging process [41]. In this context, it is important to note that the overproduction of ROSs and RNSs appears to be associated with various skin diseases due to their ability to alter protein structures, induce cellular apoptosis, and participate in proinflammatory processes [42]. For example, in fibroblasts, elevated levels of ROSs and RNSs can deactivate antioxidant systems such as catalase (CAT), glutathione peroxidase (GPx), and superoxide dismutase (SOD), leading to direct cellular damage [43]. Additionally, high concentrations of ROSs and RNSs are associated with the activation of transcription factors such as AP-1 and NF-кB, which regulate the release of signaling molecules such as proinflammatory cytokines and interleukins (IL), thus promoting skin inflammation [43,44]. On the other hand, the overproduction of ROSs also leads to an increase in matrix metalloproteinases (MMPs) levels and a decrease in tissue inhibitors of metalloproteinases (TIMP), promoting the degradation of collagen and other proteins in the ECM [3].

Table 2 summarizes the in vitro scavenging ability of cherry stem extract against O_2_^·^, NO^·^, and HOCl radicals. Table 2 also shows the IC_50_ values obtained for the EPI and GA controls. The cherry stem extract and the GA control showed no significant difference for the inhibition of the NO^·^ radical (*p* > 0.05). However, the extract did show significant differences from the EPI and GA controls for O_2_^·−^ and HOCl radical inhibition (*p* < 0.05). Despite this, the IC_50_ value of the cherry stem extract for inhibiting HOCl was close to the IC_50_ values of the EPI and GA controls. Therefore, the cherry stem extract showed high inhibition against NO^·^ and HOCl radicals and a lower inhibitory capacity against O_2_^·−^ radical. The high presence of flavonoids such as catechin, taxifolin, and chrysin in the cherry stem extract could explain these results. These compounds and their derivatives have been shown to have a high free radical scavenging capacity [45,46]. In addition, the antioxidant capacity of catechin has previously been reported in FRAP and ABTS assays [47]. The presence of flavanone naringenin and its derivatives in the extract may be related to these antioxidant results, given these compounds have previously shown a high free radical scavenging capacity, including hydroxyl radicals, superoxide, peroxide, and nitric oxide [48]. Naringenin has been associated with the inhibition of keratinocyte apoptosis and an increase in their survival [49]. 

There is growing evidence that phenolic compounds can decrease or even prevent the deterioration of the skin’s appearance and function associated with aging [50,51]. Compounds such as naringenin, quercetin, or kaempferol, noted in cherry stem extract, have shown potential beneficial effects on skin aging. The topical application of naringenin has been shown to protect hairless mice from UVB-induced skin damage. This effect was achieved through naringenin’s ability to inhibit the production of components of the senescence-associated secretory phenotype (SASP), such as the tumor necrosis factor alpha (TNF-α), IL-1β, IL-6, and IL-10, as well as lipid hydroperoxides [52]. At the same time, it was observed to maintain the expression of antioxidant genes such as GPx, glutathione reductase, and the transcription factor related to erythroid nuclear factor 2 (Nrf2) [52]. These effects were partly attributed to naringenin’s ability to reduce the levels of NF-кB, MMP-1, and MMP-3 [53]. Quercetin also proved to be an effective compound against skin oxidative stress by activating Nrf2 levels and deactivating the mitogen-activated protein kinase pathway (MAPK) and NF-кB pathways in human keratinocytes, exerting potent anti-inflammatory effects [54,55]. Recently, it was observed that kaempferol protected mice from the development of skin lesions similar to psoriasis induced by the topical administration of imiquimod. Kaempferol reduced the infiltration of CD3 T cells and the gene expression of the proinflammatory cytokines IL-6, IL-17A, and TNF-α. Additionally, it negatively regulated NF-кB signaling [56].

The results obtained were compared with those of other studies: *P. avium* leaf and stem extracts, as reported by Nunes et al., showed inhibitory activities (IC_50_) against NO^·^ of 183 and 111 mg/L, respectively. These results were significantly lower than those obtained in the present study [57]. In addition, the result for the NO^·^ uptake capacity of the cherry stem obtained by Agulló-Chazarra et al. was considerably lower than that obtained by us [26]. The antiradical values obtained in the cherry stem extract of the present study were directly related to the phytochemical richness and TPC value of the extract. However, factors such as the cherry variety, geoclimatic conditions, sample pretreatment, and extraction methods could greatly influence the composition of the extract. Therefore, it would be necessary to take these factors into account when studying the antiradical potential of cherry stems. Fábio Jesus et al. also revealed effective results of *P. avium* stem and leaf extract against the O_2_^·−^ radical [58]. Regarding other fruits of the genus *Prunus*, anti-radical studies have been carried out on *P. persica* or *domestica*, especially its peel and pulp, although no information was found on its stems or leaves. The peel and pulp extracts of these fruits showed an inhibitory activity against NO^·^ that was lower than that of the cherry stem extract and an inhibitory activity against the O_2_^·−^ radical similar to that of the cherry stem extract [59,60,61].

Considering the results obtained, cherry stem extract showed promising outcomes in terms of its ability to eliminate ROSs and RNSs. Therefore, this extract could be an excellent bioactive ingredient against oxidative stress. Its phenolic content, antioxidant capacity, and the presence of a wide variety of phenolic compounds position it as an ideal candidate for the development of cosmetic formulations that provide protection against oxidative stress.

### 2.3. Evaluation of Enzyme Inhibition Potential

As mentioned earlier, the excessive accumulation of ROSs and RNSs, influenced by both intrinsic and extrinsic factors, promotes skin aging by activating MMPs [62]. MMPs are the primary enzymes involved in ECM degradation and play a crucial role in the process of dermal renewal and remodeling. Under normal skin conditions, MMPs are expressed at minimal levels and remain inactive when bound to endogenous inhibitors, known as TIMPs. The gene expression of these enzymes can be induced in various cell types, such as keratinocytes, fibroblasts, and inflammatory cells, in response to cytokines present in the skin. IL and TNF-α, among others, are the key cytokines that activate MMP transcription [62,63,64].

An excess of reactive species can lead to harmful skin conditions by activating enzymes like collagenase, elastase, HYALase, tyrosinase, and xanthine oxidase. Collagenase and elastase, both belonging to the MMP family, degrade collagen and elastin fibers, causing wrinkles, reduced skin elasticity, and, ultimately, skin aging [65]. HYALase breaks down hyaluronic acid, essential for skin hydration, resulting in dryness and sagging [66]. Tyrosinase regulates melanin synthesis, crucial for UV protection, but its overproduction can cause skin pigmentation issues like melasma and age spots [67]. XO converts hypoxanthine to xanthine and then uric acid, but under oxidative stress, it can produce harmful superoxide radicals, contributing to various diseases and the aging process [68]. 

Numerous skin conditions are linked to these enzymatic activities, making their inhibition a promising avenue for diverse pathology treatments [69]. Studies have underscored the antioxidant potential of phenolic-rich extracts in relation to the inhibition of these proteinases [70].

Table 2 shows the results obtained for the enzyme inhibition potential of the cherry stem extract, expressed as IC_50_ values. Table 2 also shows the values of the controls used. The cherry stem extract showed the most remarkable inhibitory effect against HYALase and XO enzymes, with IC_50_ values ≤ 10 mg/L. The results were compared with standard values and the HYALase inhibition was higher with cherry stem extract than with EPI and GA, demonstrating its enzymatic potential. In addition, the cherry stem extract showed no significant difference from the EPI control in XO enzyme inhibition (*p* > 0.05). The cherry stem extract also showed potent inhibitory activity against collagenase enzyme. The result obtained by the cherry stem extract showed significant differences compared to the result obtained by the PHE control (*p* < 0.05). However, the extract showed a quite similar collagenase inhibitory potential to the PHE control. The inhibition results of cherry stem extract against tyrosinase and elastase enzymes were less remarkable, and its inhibitory activity was lower than that of the controls kojic acid and elastin, respectively.

The strong inhibitory effect of flavones and flavonols on the HYALase enzyme has been previously reported due to the presence of hydroxyl groups at the C-5 and C-7 positions of the A-ring and free hydroxyl groups at the C-3′ and C-4′ positions that enhance their inhibitory activity [71]. The presence of rutin and genistein in cherry stems may explain their high activity against the HYALase enzyme [72]. In addition, several studies have reported the anti-HYALase potential of catechin, which was characterized in cherry stems [73,74]. In the case of XO, the presence of flavonoids in cherry stems also seems to explain their inhibitory potential against this enzyme. In this context, numerous investigations have indicated that flavones and flavonols inhibit XO mainly by hydrogen bond interaction and by occupying the active site of the enzyme. The compounds pinocembrin and isoquercitrin, annotated in cherry stem extract, showed an even greater XO inhibitory effect than allopurinol [75]. Regarding collagenase, Madhan et al. reported the anti-collagenase activity of catechin through conformational changes in collagenase I from *Clostridium* histolyticum [76]. The inhibitory effect of cherry stem extract against the enzyme collagenase may be related to several mechanisms. On the one hand, the hydroxyl groups of the polyphenols interact with the collagenase skeleton, causing conformational changes that lead to the non-functioning of the enzyme [77,78]. In addition, compounds such as chlorogenic acid and hydrolysable tannins, which are metal chelators, can bind to the active center of the Zn ion of collagenase and prevent enzymatic degradation of the substrate [77,78]. On the other hand, polyphenols can intervene and protect against oxidants by modifying various biomolecular pathways in cells. For example, the protocatechuic acid present in cherry stem extract showed a high potential to induce type I collagen synthesis and inhibit MMP-1 secretion in UVA-irradiated dermal fibroblasts [79]. Catechin also significantly suppressed MMP-1 activity induced by TNF-α and prevented the inhibition of collagen synthesis in human dermal fibroblasts [80]. Similarly, caffeic acid, present in cherry stem extract in its glycosylated form, provided photoprotective effects against cytotoxicity, MMP-1 induction, and ROS formation in keratinocytes (HaCaT) [81]. Given the wide range of phenolic compounds present in the analyzed extract, including rutin, genistein, pinocembrin, isoquercitrin, catechin, protocatechuic acid, and caffeic acid, it can be concluded that cherry stem extract exhibits significant anti-collagenase activity.

Demi et al. studied the anti-inflammatory activity of cherry stem extract and evaluated its ability to inhibit the XO enzyme, obtaining an IC_50_ of 4720 mg/L of extract, showing a much lower inhibition than in the present study [24]. Agulló-Chazarra et al. evaluated cherry stem extract and it did not show any activity against collagenase enzyme inhibition [26]. Furthermore, Babota et al. evaluated the ability of cherry stem extract to inhibit tyrosinase enzyme, obtaining an IC_50_ value of 8.66 mg/mL and showing, therefore, a lower inhibition [33]. The enzyme values reported in these studies may have been lower due to the lower phytochemical richness of the extracts used compared to the extract of the present study, whose phytochemical profile was very rich.

Therefore, all these results could point to the cherry stem as an important source of different anti-inflammatory and antioxidant compounds.

### 2.4. Assessment of the Stability of the Cherry Stem Extract

The cherry stem extract, previously diluted, was subjected to extreme conditions, including centrifugation, a temperature ramp, and heating-cooling cycles, with the purpose of independently evaluating its stability prior to its incorporation into a cosmetic formulation. After these stability tests, a decrease in the TPC values of the extract was observed, being more pronounced following the temperature ramp and the cycles (Figure 2). Initially, the TPC value of the extract was 131.40 mg GAE/g of sample, while after centrifugation, the temperature ramp, and the cycles, the TPC values were 103.03, 77.72, and 56.11 mg GAE/g of sample, respectively. During the stability tests, the extract proved to be sensitive to temperatures up to 80 °C and sudden temperature changes (4 °C and 60 °C), resulting in a significant decrease in its TPC values. This decrease may be attributed to the high temperatures, which negatively affect the bioactive compounds present in the extract, leading to their degradation.

The cherry stem extract stands out as a significant source of phenolic compounds. However, these tests have revealed its high susceptibility to high-temperature conditions. In this regard, previous research has indicated that elevated temperatures can trigger the degradation of phenolic compounds present in plants and herbs [82,83]. For example, flavonoids like catechin and epicatechin can undergo complex oxidation and polymerization reactions when exposed to high temperatures [84]. Therefore, the application of the extract will be limited by its instability under different storage conditions. It is crucial to establish optimal storage conditions and develop cosmetic formulations that promote its stability in order to preserve its antioxidant potential and potential benefits, especially for skin health.

### 2.5. Development of a Cosmetic Formulation and Study of Its Stability

The science of cosmetic formulation has always been a challenge due to the importance of skin care. Many factors such as the pH, viscosity, and ingredients, among others, have to be considered in the development of a cosmetic product [51].

In addition, there is currently a strong demand for cosmetics with natural ingredients, which has become a serious challenge for the cosmetics industry [8]. Polyphenols have gained popularity as active ingredients in cosmetic formulations due to their beneficial effects on the skin [5]. In this regard, the antioxidant power and phytochemical profile of cherry stem extract made it an ideal candidate for incorporation into a cosmetic formulation and for study of the stability of the formulation.

In the present study, a cosmetic gel formulation was developed with additions of 0.5%, 1%, and 2% cherry stem extract. A gel without extract was also developed and used as a control. The gels, which presented good organoleptic conditions, were subjected to different stability tests. In addition, the physicochemical parameters pH, viscosity, and color were evaluated to ensure the stability and viability of the formulation. The TPC value of the gels was also determined during the stability tests in order to observe if there was a possible degradation of the compounds present in the extract.

#### 2.5.1. Assessment of Initial Physicochemical Characteristics and Total Phenolic Content (TPC) in Formulations

The physicochemical characteristics, including pH, viscosity, and color, were assessed for the gel samples comprising no extract (0%) as well as containing cherry stem extract (0.5%, 1%, and 2%). Furthermore, the TPC was quantified using the Folin–Ciocalteu method for all formulations. Table 3 presents the initial measurements of all of these parameters.

Skin pH is an important factor to consider when designing a cosmetic formulation. Numerous studies report varying skin pH values, with a pH range from 4 to 7, although the natural pH of the skin surface is estimated to be 4.7 [85,86,87]. Therefore, the pH of the cosmetic product to be developed must be similar to that of the area of application on the skin. The pH also plays an important role in the preservation of a cosmetic product. An acidic pH contributes to the antimicrobial action of these products [88]. Table 3 shows the initial pH values of the formulations at different extract concentrations. The gel without extract had a pH of 5.92, which decreased with an increasing extract concentration. The pH values for the gels with 0.5%, 1%, and 2% extract were 5.37, 4.96, and 4.66, respectively.

An important factor when developing a topical formulation is viscosity. The viscosity of a cosmetic product determines its consistency and spreadability on the skin and is a significant factor in the quality and stability of a cosmetic [89]. In this sense, cosmetic formulations often have a pseudoplastic behaviour that favors the application and spreading of the product on the skin [90,91]. Therefore, rheological properties are one of the most important parameters to measure as they will also act as an indication of changes in the formula [92]. In this regard, gels have become increasingly used for topical application due to their numerous advantages. Some of the useful advantages of gels in this field are their structure, which significantly improves the stability of the product, and their macroscopic properties such as texture and spreadability, which make them well accepted [93,94]. However, for a cosmetic formulation to be considered a gel, its viscosity must be between 1 and 100 Pa·s and the thickening agents in the gel must be less than 10%. In Table 3, we can observe the initial viscosities of the gels. The viscosity of the gels increased with an increasing extract concentration; however, all the gels showed adequate viscosity within a range from 1 to 100 Pa·s.

Other parameters to highlight in cosmetic formulation science are organoleptic parameters such as color. Color is one of the most relevant attributes in cosmetic products, and color changes in the product can indicate the loss of stability or degradation of the ingredients used. Furthermore, the evaluation of color in cosmetics can be a complementary test to determine consumer acceptance [92,95]. For the evaluation of color in cosmetic products, several researchers have used the CIELab space [96,97]. This space is characterized by three color coordinates: L*, a*, and b* [98]. The L* parameter represents brightness and is aligned with the vertical axis, ranging from 0 to 100, where L* = 0 signifies complete opacity, and L* = 100 denotes full transparency. On the other hand, the parameters a* and b* correspond to the perpendicular horizontal axes, delineating the spectrum from red to green and from blue to yellow, respectively. The a* parameter spans the range between the colors red and green, with values ranging from −100 to +100. Positive a* values indicate a reddish hue, while negative values signify a greenish tint [98]. The b* parameter encompasses the spectrum between yellow and blue, with values ranging from −100 to +100. Positive values of b* indicate a yellow color while negative values of b* indicate a blue color [98]. As for the color of the gels, the lightness parameter L* decreased in the gels with an increasing extract concentration. Negative values of a* indicated the presence of green shades. After the addition of the extract, there was a very small increase in the parameter a* with the appearance of reddish tones. The parameter b* showed positive values, indicating the prevalence of a yellow color in the gels. After the addition of the extract, there was an increase of about 5 units in the b* values.

As for the phenolic content, the highest TPC value was observed in the gel containing 2% extract. As expected, the results of the TPC values showed an increase in phenolic content as the concentration of cherry stem extract in the gels increased.

#### 2.5.2. Cosmetic Formulation Stability Study

Stability assessment of a cosmetic product is an essential requirement to ensure the safety of a cosmetic product. The main purpose of stability studies is to determine the expiry date of a cosmetic product, provided it has been stored correctly [99].

In the present work, the accelerated stability was evaluated for all formulations by centrifugation, temperature ramp, and heating-cooling cycles. In addition, the long-term stability was evaluated under light and dark conditions for three months.

##### Accelerated Formulation Stability

Accelerated or preliminary stability studies can guide the manufacturer in the pre-formulation process and allow them to obtain information about the physicochemical changes that may occur in the cosmetic when it is exposed to extreme conditions [100].

In this regard, the accelerated stability tests allowed us to verify the thermal stability of the formulations, both at high and low temperatures, even for the formulation containing 2% extract. The formulations remained intact in all tests, preserving their initial appearance and homogeneity, except in the cycling test where small alterations were observed. In addition, the formula without extract showed the most variations throughout the stability study, so the stability of the other gels may be due in part to the incorporation of the cherry stem extract and the presence of antioxidant compounds. According to Darvin et al., antioxidant compounds, such as flavonoids, are used by cosmetic companies to protect formulations against free radical-induced oxidation and may increase the shelf life of cosmetic formulations [101]. Therefore, the presence of cherry stem extract, rich in phenolic compounds, could have contributed to the protection of the formulations during the stability studies. In addition, small variations at elevated temperatures or cycles are acceptable when carrying out a stability study on a cosmetic product [92].

The pH values showed hardly any variation in the different stability tests, which could indicate that the behaviour of the formulation’s ingredients remained practically unchanged. The pH stability could in turn indicate chemical stability and microbiological stability [102]. Figure 3 shows the pH values during the accelerated stability tests. The pH values ranged between 4.53 and 5.93, indicating a pH compatible with the skin, considering that the pH gradient of the skin surface varies between 4 and 7 [87].

The viscosity of the gels remained stable after the accelerated stability tests. All gels maintained a homogeneous appearance and no loss of integrity was observed at any time. Figure 4 shows the viscosity values after the accelerated stability tests. The viscosity of the gels decreased mainly after the centrifugation test. This decrease may be related to the pseudoplastic behaviour of the gels, whose viscosity decreases with an increasing shear rate [103]. This rheological behaviour is characteristic of gels and is positively valued, as it indicates that they can be easily spread on the skin [104]. During ramping and temperature cycling, the viscosity evolved in a similar manner. Despite this, the gels presented a viscosity between 7.4 and 15.3 Pa·s at 10 rpm, being within the optimal viscosity range for a gel formulation.

Figure 5 summarizes the color parameters during the accelerated stability. The L*, a*, and b* values showed a variation of less than five units from the zero-time values. We can observe how the heating-cooling cycles were the process that most affected the color of the gels due to the abrupt temperature changes. The color variations could be related to the degradation of the bioactive ingredient due to the exposure of the gels to high temperatures. Therefore, it is possible that oxidation of the phenolic compounds present in the cherry stem extract was the cause of the color change [30,105]. Despite this, the gels showed good organoleptic characteristics and a homogeneous appearance.

Figure 6 displays the TPC values obtained after the accelerated stability tests. The gels containing 0.5% and 1% cherry stem extract in their formulation showed consistent TPC values, suggesting stability under temperature conditions. This stability could be attributed to the inclusion of the extract in the cosmetic formulation, which has provided protection to the phenolic compounds present in it. However, the formulation incorporating 2% extract showed variations in its TPC values. These variations were observed mainly after the temperature ramp and centrifugation test. Specifically, a decrease was noted after centrifugation, which could be due to the precipitation of bioactive compounds. In the case of the temperature ramp, these fluctuations could be attributed to the gel’s exposure to high temperatures. Interestingly, during the temperature ramp, an increase in the TPC value of the gel with 2% extract was observed, possibly due to the release of glycosides or phenolics bound in free phenolic derivatives during this process [106]. However, further studies are required for a more precise understanding of this phenomenon.

There are currently very few studies that report detailed results of accelerated stability tests of cosmetic formulations that have incorporated plant extracts. The results obtained from the accelerated stability tests were compared with other studies focused on other type of extracts: Vieira et al. observed a pH reduction in a facial mask formulation developed with fermented soy milk, where they obtained pH alterations of greater magnitude under storage conditions at 45 °C, which were accepted and considered as stable [107]. Ferreira et al. also developed a topical formulation with avocado peel and, after evaluating the accelerated stability, observed changes in the viscosity of the formulation [108].

The results of the accelerated stability assessment indicate that the developed formulations demonstrated good stability. The most significant variations were observed during the temperature cycling test. Given that minor fluctuations are considered acceptable for cosmetic products exposed to extreme temperatures [92], we can conclude that the formulations developed in this study maintained their stability during the accelerated stability assessment. It is important to note that these formulations provided protection to the extract, helping to partially prevent the degradation of bioactive compounds, and maintaining constant TPC values in most cases. Additionally, optimal physicochemical values for skin application were observed. However, it is crucial to continue researching and advancing the development of new stable cosmetics that incorporate bioactive extracts.

##### Long-Term Formulation Stability

During the long-term stability test, the pH, viscosity, color, and TPC values of the gels were evaluated at time zero (T0) and once a month for three months (T1, T2, and T3) during light and dark storage.

No pH variations affecting the stability of the gels were observed. Figure 7 shows the pH evolution of the gels stored in the light and in the dark. The pH values were constant in the gels containing the cherry stem extract. However, the gel containing no extract showed a greater variation in pH values. These results could be due to the presence of the extract, which conferred greater stability to the gels incorporating it. Due to its antioxidant properties, the cherry stem extract could have provided greater protection to the formulation. It should be noted that the pH of all of the gels remained between 4 and 7, making them compatible with the pH of the skin and suited for topical use.

The long-term stability of the gels was characterized by the absence of sedimentation and the maintenance of gel integrity. Figure 8 shows the viscosity values obtained during the stability study. The viscosity of the gels remained practically constant over time. However, in the third month there was a small decrease in viscosity which did not affect the quality of the gels. This decrease in viscosity could be due to changes in ambient temperature during the months of the study. However, the viscosities of the gels remained between 1 and 100 Pa·s, a range in which a cosmetic is considered a gel.

Figure 9 shows the L*, a*, and b* values of the gels during the storage time in the light and dark. The L* parameter remained constant over time with an average variation of 1.47 units in the light and 0.97 units in the dark. The a* parameter showed slight variations, mainly at the beginning of the study. All gels had a similar behaviour in terms of a* values. Positive a* values were observed, indicating the presence of reddish tones. However, the average variation in the light and dark conditions was 0.69 and 0.55 units, respectively. The b* parameter decreased in all gels, which was associated with a loss of yellow color. Formulations stored in the dark showed smaller variations in the b* parameter than those stored in the light. However, the average variation was 0.73 units in the light and 0.62 units in the dark. Considering the CIELab scale, the L*, a*, and b* parameters remained stable over time, both in the light and in the dark.

Regarding the phenolic content of the gels, Figure 10 shows the TPC values of the gels stored in light and darkness. The TPC values decreased slightly during the first two months of storage. However, from the second month onwards, the values remained constant, except for the 0.5% formulation stored in light, where the TPC value continued to decrease until T3. This reduction could primarily be linked to the degradation of bioactive compounds due to factors such as the storage time, exposure to light, or temperature. Considering that, previously, the diluted extract experienced a significant decrease in TPC values under various temperature conditions, in this case, the variation in the TPC values was minimal. Therefore, we can infer that the inclusion of the extract in the cosmetic formulation favored its stability. Despite the slight variations, the formulations continued to display acceptable TPC values until the end of the study.

The long-term stability of the formulations was compared with other similar studies. Huma et al. also observed a decrease in pH after 72 days of study of a formulation developed with *Beta vulgaris* leaf extract as an active ingredient, where they obtained pH variations of greater magnitude, which was considered as stable [109]. Pinto et al. considered as stable a topical formulation with chestnut shell extract that, after 30 days of storage at 25 °C, showed greater changes in color parameters than the one developed with the present cherry extract after 90 days of storage. Rodrigues et al. evaluated the stability of a topical formulation with coffee silverskin extract and reported that, after 180 days of storage, the pH, viscosity, L*, a*, and b* parameters of the formulation varied similarly to the results reported in this work after 90 days [110]. In addition, Rodrigues et al. also observed a decrease in the phenolic content of a topical formulation with coffee silverskin extract during a stability test of 180 days [111].

The results of the long-term stability tests indicated that formulations containing cherry stem extract remained stable for three months, both under light and dark conditions. Minimal variations in physicochemical parameters were observed, and no sedimentation or gel breakage issues were detected in any of the formulations. In summary, the tests demonstrated that the studied formulations are stable for a period of three months in illuminated and dark environments and exhibit suitable characteristics for topical use on the skin.

## 3. Materials and Methods

### 3.1. Chemical and Reagents

For HPLC analysis, water, formic acid, and acetonitrile, which were of HPLC-MS grade and used as received, were provided from Fluka (Sigma-Aldrich, Steinheim, Germany) and Lab-Scan (Gliwice, Sowinskiego, Poland), respectively. Regarding extraction processes and solutions, we used milli-Q Millipore (Bedford, MA, USA) ultrapure water and absolute ethanol, which were purchased from VWR Chemicals (Radnor, PA, USA).

All reagents for TPC, antioxidant activity, radical scavenging, and enzyme inhibition experiments were sourced from Sigma-Aldrich (St. Louis, MO, USA) except NOC-5 which was obtained from Chemcruz (Santa Cruz Biotech., Dallas, TX, USA). For cosmetic prototypes, citric acid, xanthan gum, and glycerin were purchased from Guinama S.L.U (Valencia, Spain).

### 3.2. Sample Preparation

The cherry stems were kindly donated by the company ‘‘Cerezas y Almendras Castillo S.L.’’ located in Castillo de Locubín, Jaén. The stems were dried in the sun for 24 h. Once dried, they were crushed to reduce their granulometry to below 1 mm in order to facilitate the extraction process.

To obtain the cherry stem extract, the same methodology was used as in García et al. [17]. The extracts obtained from the cherry stem had a caramel-like appearance and texture with a dark brown shade.

### 3.3. Analysis HPLC-ESI-qTOF-MS/MS

For the determination of the chemical profile of the cherry stem extract, an HPLC-ESI-qTOF-MS analysis was conducted on an HPLC 1290 platform, with a Jet Stream dual electrospray ionization (ESI) interface, coupled to a quadrupole time-of-flight (qTOF) mass spectrometer (G6530C UHD, Agilent Tech., Santa Clara, CA, USA). 

Initially, the cherry stem extract was reconstituted to a concentration of 5 mg/mL. Subsequently, the solution underwent centrifugation and filtration (0.2 µm), and was then transferred to an HPLC vial for analysis. Additionally, blank samples were prepared to detect any potential contaminants. 

The chromatographic and mass spectrometry conditions and parameters were applied following the previously optimized method [17].

The acquired raw data underwent initial transformation using the MSConverGUI software v3.0.21054-976edd878 (https://proteowizard.sourceforge.io/download.html, accessed on 28 September 2023) and were further processed in MZmine 2.53. This encompassed multiple data processing steps, including background noise detection, ADAP chromatogram builder, ADAP chromatogram deconvolution, isotope grouping, and alignment. The specific parameters used in data processing were reported previously [17]. Finally, the prediction of molecular formular and chemical structures was performed with Sirius 5.8.1. The information obtained from Sirius was compared with various databases (Human Metabolome Database (HMDB), MassBank of North America, CEU mass mediator, among others) in order to carry out the annotation of the compounds. The compounds were classified into 4 levels. Level 1 was achieved by using commercial standards, level 2 was achieved by comparing MS/MS with those available in the databases, level 3 focused on molecular formula and comparison of MS1 spectra, and level 4 refers to all those signals that could not be annotated and were therefore unknown [112].

### 3.4. In Vitro Antioxidant Activity

For the evaluation of the antioxidant power of cherry stem extract, the assays were performed by measuring absorbance in a Synergy H1 Monochromator-Based Multi-Mode Micro plate reader (Bio-Tek Instruments Inc., Winooski, VT, USA) equipped with a thermostat for fluorescence and UB/Vis measurements. All assays were carried out following previous studies.

#### 3.4.1. Evaluation of Antioxidant and Antiradical Activities

Total phenol content (TPC) was determined by the Folin–Ciocalteu method. The results were expressed as milligrams of gallic acid equivalents (GAE) per gram of dry extract (DE). In addition, the antioxidant capacity of the cherry stem was evaluated using the FRAP, TEAC, and ORAC assays. The results were expressed in the FRAP method as mmol FeSO_4_ equivalents per gram DE (mmol FeSO_4_/g DE). In the TEAC method, the results were expressed as μmol Trolox equivalents per gram DE (μM Trolox/g DE) and in the ORAC method as mmol Trolox equivalents per gram DE (mmol Trolox/g DE). All these methods have been previously described in several studies and have been carried out with slight modifications [113,114,115,116]. All measurements were performed in triplicate.

The ability of cherry stem extract to inhibit superoxide radical (O_2_^·−^), hypochlorous acid (HOCl), and nitric oxide (NO^·^) was evaluated. The previously described methodology was followed but with slight modifications [117]. The scavenging properties of cherry stem extract and positive controls against the O_2_^·−^ radical were assessed colorimetrically by reduction of NBT to diformazan purple. The ability of the cherry stem extract to inhibit HOCl by inducing the oxidation of DHR to rhodamine was determined. The scavenging activity of cherry stem extract against NO^·^ was determined by oxidation of DAF-2. To assess the inhibition of O_2_^·−^, NO^·^, and HOCl radicals, epicatechin (EPI) and gallic acid (GA) were employed as reference controls. All results were expressed as IC_50_ value using different concentrations of cherry stem extract and all measurements were performed in triplicate.

#### 3.4.2. Evaluation of Enzyme Inhibition Potential

The potential of cherry stem extract to inhibit hyaluronidase (HYALase), collagenase, elastase, tyrosinase, and xanthine oxidase enzymes was evaluated. The methods were carried out following previous studies [118,119,120].

The HYALase inhibition assay involves quantifying the reduction in transmitted light intensity caused by particles generated from the breakdown of hyaluronic acid (HYAL) by the HYALase enzyme. Absorbance readings were taken at 600 nm. EPI and GA were employed as controls to evaluate HYALase inhibition. The outcomes from the EPI and GA controls were reported as IC_20_ values.

The evaluation of collagenase enzyme activity is based on the colorimetric analysis of the degradation of the FALGPA substrate by the enzyme. The absorbance measurements were taken at 335 nm. Phenanthroline (PHE) was used as the control to assess collagenase inhibition, and the IC_50_ value was quantified as the control.

The assessment of elastase enzyme inhibition was based on the release of p-nitroaniline resulting from the hydrolysis of the substrate MeOSuc-Ala-Ala-Pro-Val-pNa by elastase. The quantity of p-nitroaniline was gauged through absorbance readings at 405 nm. Elastatinal (ELA) was employed as the control for elastase inhibition. The control result was stated as % inhibition.

The evaluation of tyrosinase inhibition was conducted using a “Colorimetric Tyrosinase Inhibitor Detection Kit” sourced from Sigma-Aldrich, USA. Inhibition control was established with kojic acid (KA). The control outcome was represented as % inhibition.

The XO inhibition assay was carried out using the “Cayman’s Xanthine Oxidase Fluorometric Assay Kit” from Cayman Chem. (Ann Arbor, MI, USA). EPI was employed as the control for XO inhibition, and the resulting IC_50_ value was noted as the control outcome.

Results for cherry stem extract were presented as IC_50_ values at various extract concentrations, except for the enzyme tyrosinase, which was presented as IC_55_ value at various extract concentrations. All evaluations were performed in triplicate.

### 3.5. Assessment of the Stability of the Cherry Stem Extract

Accelerated stability tests were conducted to assess the stability of the cherry stem extract. The extract was diluted in an ethanol and water solution at 80:20 (*v*/*v*) and then subjected to centrifugation, temperature ramp, and heating-cooling cycles. Centrifugation was performed using a Rotofix 32 A centrifuge (Hettich Zentrifugen, Tuttlingen, Germany) at 3000 rpm and room temperature for 30 min. During the temperature ramp, the temperature gradually increased from 20 °C to 80 °C, with increments of 5 °C every 30 min. The heating-cooling cycles maintained the extract at 60 °C and 4 °C, respectively, for 24 h each. A total of six cycles were performed. The stability of the extract was evaluated by measuring the TPC value using the Folin–Ciocalteu method before and after the stability tests. Concentration sweep was conducted to determine the TPC value. All measurements were performed in triplicate.

### 3.6. Incorporation of the Extract in a Cosmetic Formulation and Study of Its Stability

#### 3.6.1. Cosmetic Formulation

A cosmetic formula was developed in the form of a gel using the Unguator EMP (Microcaya, Vizcaya, Spain). The ingredients used were water, xanthan gum, glycerine, ethanol, citric acid, and cherry stem extract. Cherry stem extract was added to the formula in different percentages (0.5%, 1%, and 2%). Table 4 presents a comprehensive list of ingredients, including their corresponding INCI names, CAS numbers, and cosmetic functions. The gel containing no extract was used as a control. Three sets were assembled, each comprising four gels with varying extract percentages. One set was designated for accelerated stability testing, while the remaining two were allocated for long-term stability assessments.

To prepare the gel, the xanthan gum was initially hydrated. This entailed gradually dispersing the xanthan gum over water at 60 °C with continuous stirring using a magnetic stirrer. Once well-mixed, ethanol and glycerine were added to the xanthan gum–water mixture, and stirring continued until a uniform mixture without lumps was achieved. Separately, citric acid was dissolved in water. This citric acid solution was then added to the thoroughly mixed gel to ensure a consistent blend without lumps. The blend of ingredients was subsequently transferred to the Unguator to complete the gel formation. In the Unguator, gel formation took place in two phases: the first phase, known as the wetting phase, involved mixing the ingredients at 2150 rpm for 30 s. The second phase, referred to as the swelling phase, entailed mixing at 600 rpm for 9 min and 30 s. The same procedure was followed for incorporating the cherry stem extract into the gel. Before incorporating, the extract was dissolved in a blend of water and ethanol, then amalgamated with the remaining ingredients and processed using the Unguator. The water content was adjusted according to the extract concentration. The pH of the gel was adjusted by adding citric acid. Ethanol played a vital role in improving the stability and extending the shelf life of the cosmetic formulation [121,122].

#### 3.6.2. Study of Stability

Accelerated stability tests assessed the stability of gels under harsh conditions. To carry out the accelerated stability tests, the ISO/TR 18811 ‘‘Cosmetics—Guidelines on the stability testing of cosmetic products’’ and the previously described methodology were followed [17,123]. For this purpose, accelerated stability was evaluated by centrifugation, temperature ramp, and heating-cooling cycles. Centrifugation was conducted with a Rotofix 32 A centrifuge (Hettich Zentrifugen, Tuttlingen, Germany) at 3000 rpm and room temperature for 30 min. The temperature ramp was evaluated by a progressive increase in the temperature from an initial temperature of 20 °C to a final temperature of 80 °C. The temperature was increased by 5 °C every 30 min. The heating-cooling cycles were carried out for 24 h at 60 °C and 4 °C, respectively. A total of six cycles were performed. The physicochemical parameters pH, viscosity, color, and TPC were evaluated after each method.

Long-term stability was assessed over a three-month period. The gels were stored for three months, both exposed to light and kept in the dark, at room temperature. The physicochemical parameters pH, viscosity, color, and TPC were evaluated once a month.

For pH measurement, 1 mL of gel was extracted and subsequently diluted with water in a 1:10 (*v*:*v*) ratio at room temperature. This procedure utilized a Fisher-brand™ Accumet™ Portable AP110 pH meter (Thermo Fisher Scientific, Waltham, MA, USA). Viscosity measurements were conducted using an IKA ROTAVISC me-vi Complete viscometer (IKA Designed for Scientist, Staufen, Germany). The 11SP spindle was used at a speed of 10 and 70 rpm. The viscosity values were recorded after 60 s and expressed as Pa·s. The color of the gels was assessed using a Lovibond TR 500/520 Series colorimeter (Lovibond Tintometer Group, Amesbury, UK). The color parameters L*, a*, and b* were determined. L* indicated the brightness of the gels, a* indicated the coordinates for red/green. b* indicated the coordinates for yellow/blue. The TPC of the gels was assessed using the Folin–Ciocalteu method described above, but with minor modifications. All measurements were carried out in triplicate and at room temperature.

The physicochemical criteria set to assess the stability of the gel formulation were as follows: a viscosity within a range from 1 to 100 Pa·s, which is indicative of a gel consistency, and a pH level falling between 4 and 7, ensuring suitability for skin application [86,124].

### 3.7. Statistical Analysis

Differences among formulations were evaluated through one-way analysis of variance (ANOVA). Statistical significance for variations between pairs of data was considered significant at *p* < 0.05. The analysis was performed using IBM’s Statistical Package for the Social Sciences (SPSS 15.0).

## 4. Conclusions

In this study, a total of 146 compounds were tentatively annotated by HPLC-ESI-qTOF-MS/MS in cherry stem extract. The most representative groups were organic acids, phenolic acids, flavonoids, glycosylated flavonoids, and procyanidins. Sixty-two compounds were annotated for the first time in cherry stem extract. The cherry stem extract exhibited notable antioxidant effects, specifically in its electron-donation capacity (FRAP), hydrogen atom transfer (ORAC), and inhibition of radical species (O_2_^·−^, NO^·^, and HOCl). Additionally, the cherry stem extract demonstrated effective inhibition of HYALase, collagenase, and XO enzymes. The stability of the extract was evaluated by subjecting it to different temperature conditions. During the tests, it was observed that the diluted extract was sensitive to high temperatures, resulting in a decrease in the TPC values.

The extract was integrated into a cosmetic formulation with favorable organoleptic and physicochemical properties. The formulation’s stability was verified through accelerated stability testing and long-term stability testing. The cosmetic formulation was stable during the stability testing and showed physicochemical properties compatible with skin and for topical use. Contrary to the TPC values of the diluted extract, the TPC values of the formulations hardly changed during the stability tests, especially during the long-term stability testing. In summary, the developed formulations not only demonstrated stability but also provided protection to the extract. Through this preliminary study, it was observed that cherry stems could be a promising source of bioactive ingredients for skin care and for the possible development of safe and effective cosmeceuticals. In the future, it would be interesting to further evaluate the beneficial properties of cherry stem extract on the skin and to study the development of new cosmetic formulations incorporating it as a bioactive ingredient for further in vivo study.

## Figures and Tables

**Figure 1 ijms-25-03723-f001:**
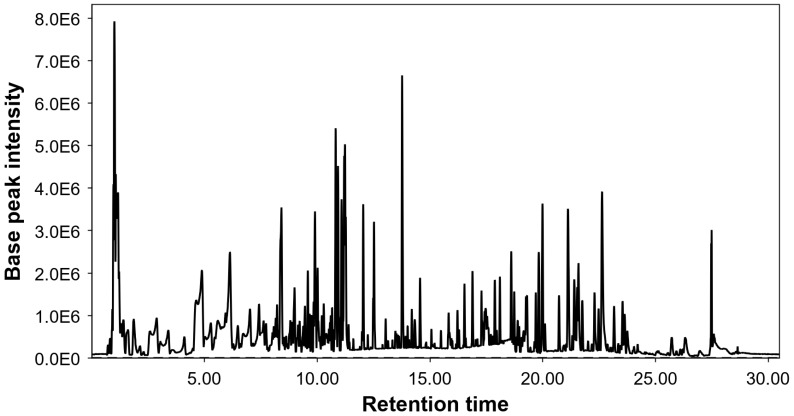
Base peak chromatogram of cherry stem extract. The chromatogram of the cherry stem extract is included in the Appendix A, enlarged and numbered for better visualization of the individual chromatographic peaks. Due to the sensitivity of the HPLC-ESI-qTOF-MS technique performed and the fragments obtained, a detailed characterization of the cherry stem extract could be carried out.

**Figure 2 ijms-25-03723-f002:**
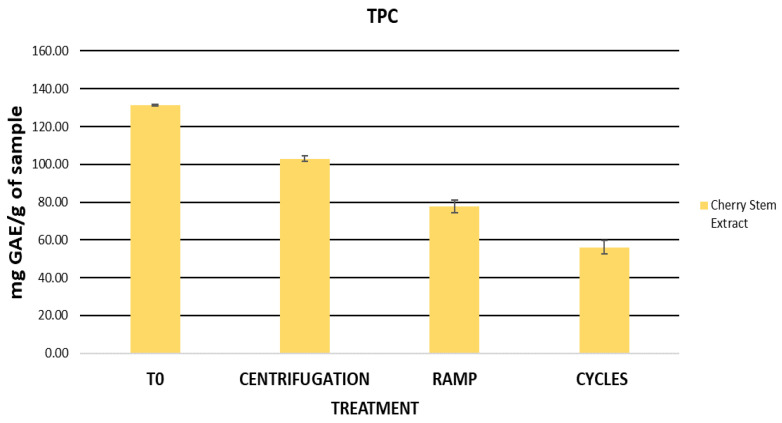
TPC values during accelerated stability of the extract. GAE: gallic acid equivalent. The error bars represent the standard deviation of the triplicates of the analytical measurements.

**Figure 3 ijms-25-03723-f003:**
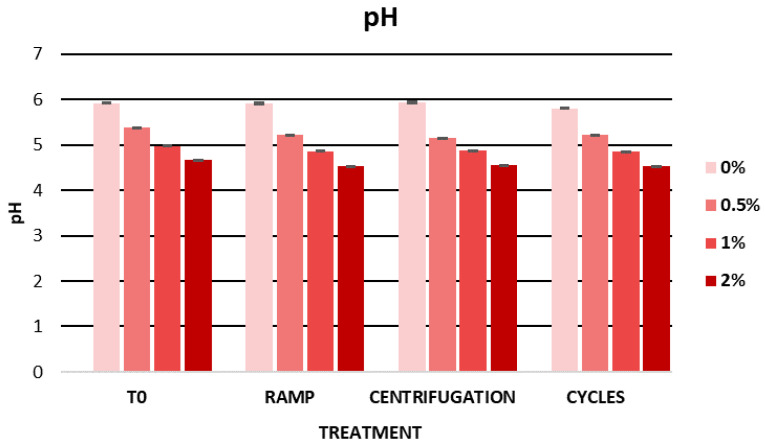
pH values during accelerated stability. The 0%, 0.5%, 1%, and 2% values mean the different percentages of extract used for the gel formulations. The error bars represent the standard deviation of the triplicates of the analytical measurements.

**Figure 4 ijms-25-03723-f004:**
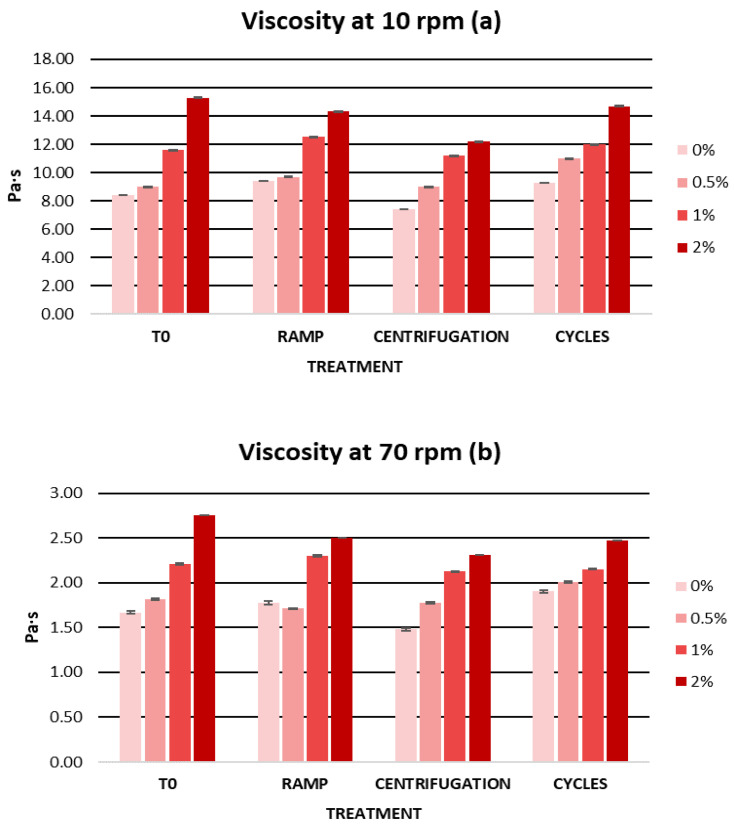
(**a**) Viscosity values at 10 rpm during accelerated stability; (**b**) viscosity values at 70 rpm during accelerated stability. rpm: revolutions per minute. The 0%, 0.5%, 1%, and 2% values mean the different percentages of extract used for the gel formulations. The error bars represent the standard deviation of the triplicates of the analytical measurements.

**Figure 5 ijms-25-03723-f005:**
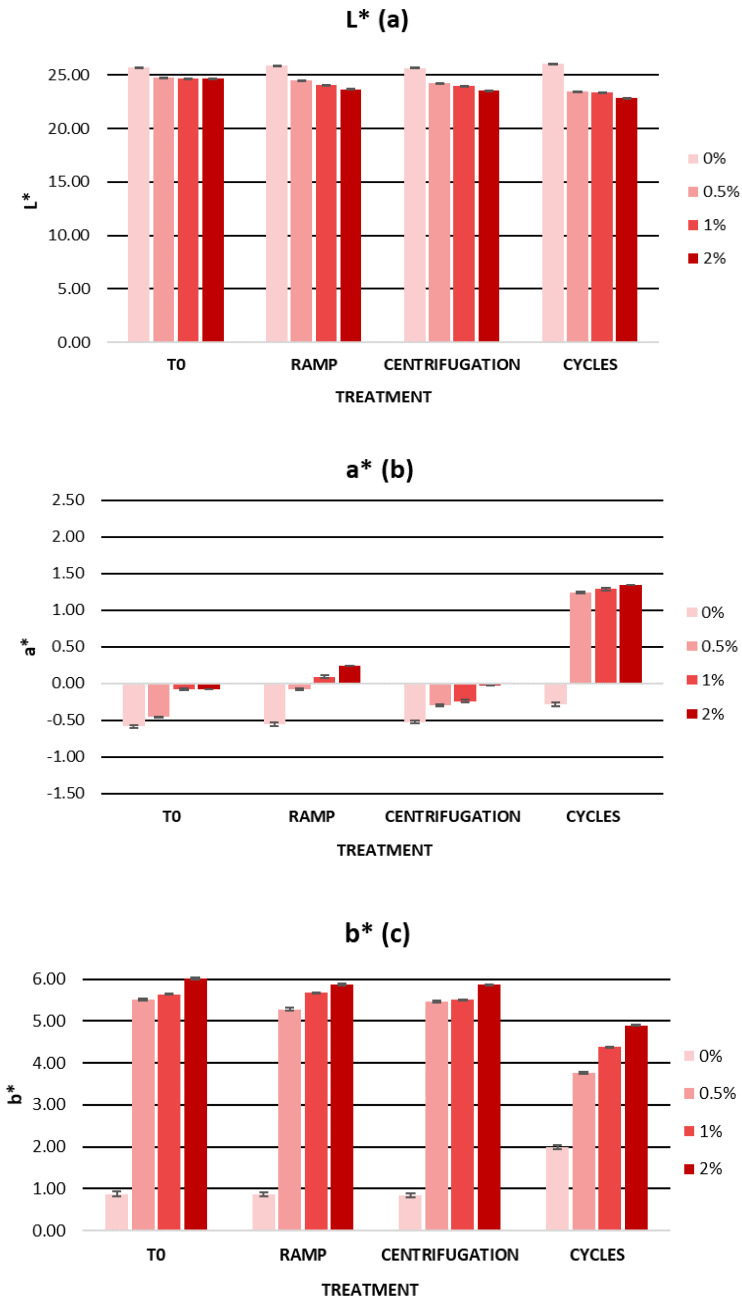
(**a**) Parameter L*; (**b**) parameter a*; (**c**) parameter b* during accelerated stability. The 0%, 0.5%, 1%, and 2% values mean the different percentages of extract used for the gel formulations. The error bars represent the standard deviation of the triplicates of the analytical measurements.

**Figure 6 ijms-25-03723-f006:**
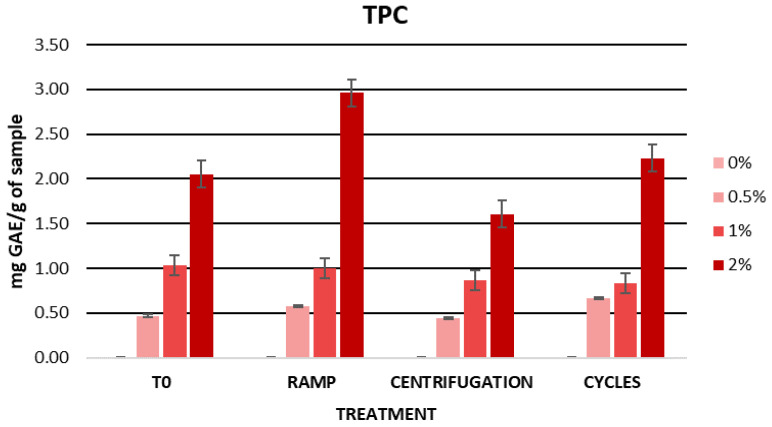
TPC values during accelerated stability. GAE: gallic acid equivalent. The 0%, 0.5%, 1%, and 2% values mean the different percentages of extract used for the gel formulations. The error bars represent the standard deviation of the triplicates of the analytical measurements.

**Figure 7 ijms-25-03723-f007:**
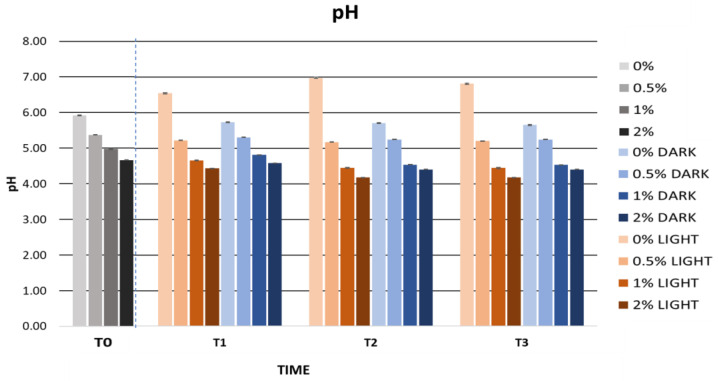
pH values during long-term stability testing. The 0%, 0.5%, 1%, and 2% values mean the different percentages of extract used for the gel formulations. DARK: samples stored in the dark; LIGHT: samples stored in the light. The dotted line marks the boundary between the control group and the samples during long-term stability testing. The error bars represent the standard deviation of the triplicates of the analytical measurements.

**Figure 8 ijms-25-03723-f008:**
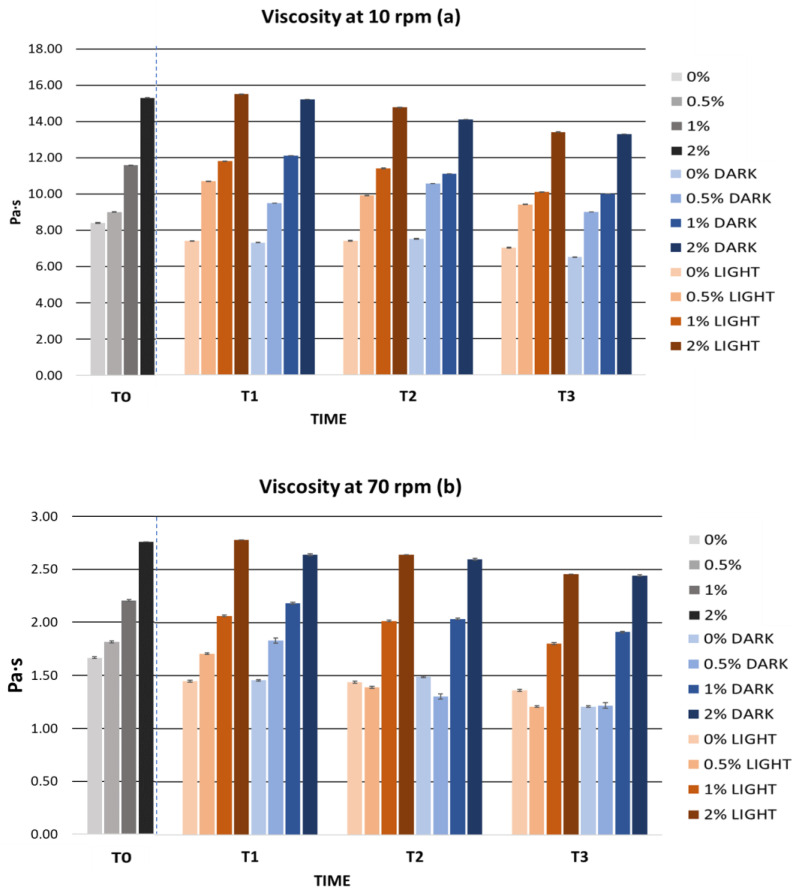
(**a**) viscosity values at 10 rpm during long-term stability test; (**b**) viscosity values at 70 rpm during long-term stability test. rpm: revolutions per minute. The 0%, 0.5%, 1%, and 2% values mean the different percentages of extract used for the gel formulations. DARK: samples stored in the dark; LIGHT: samples stored in the light. The dotted line marks the boundary between the control group and the samples during long-term stability test. The error bars represent the standard deviation of the triplicates of the analytical measurements.

**Figure 9 ijms-25-03723-f009:**
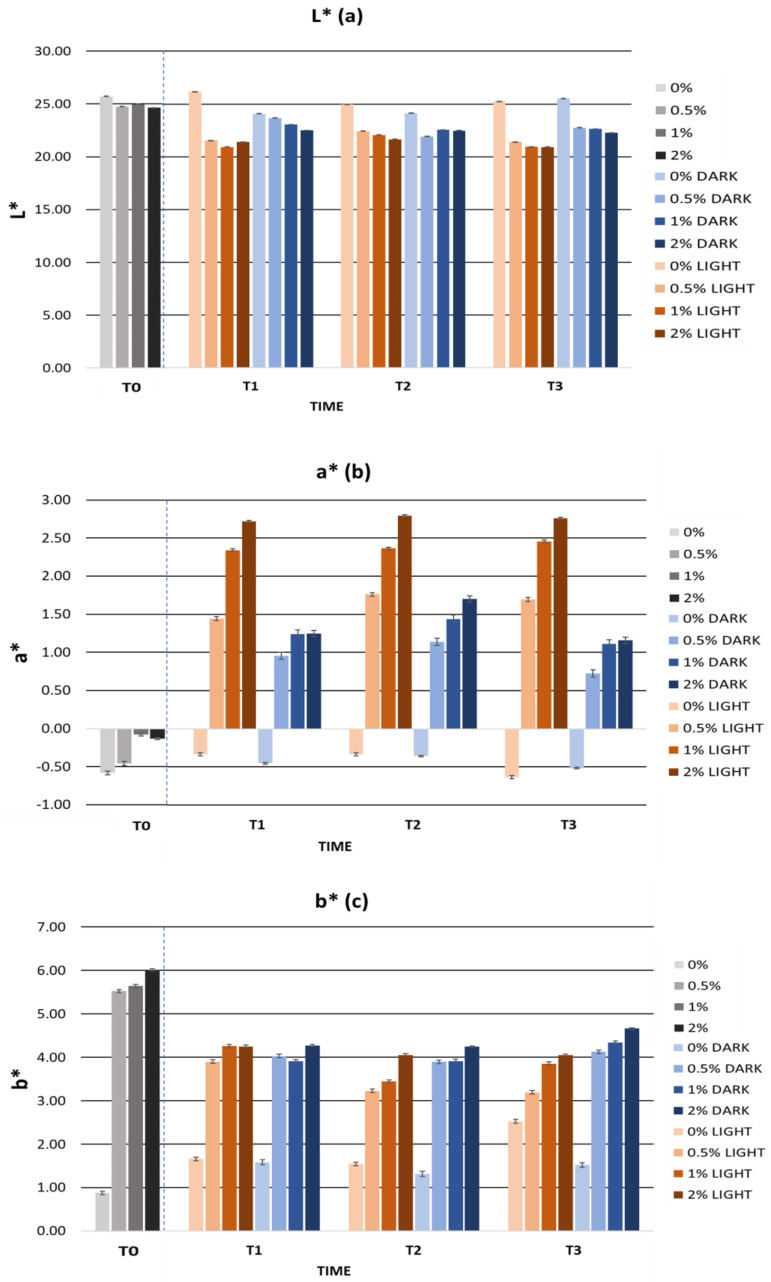
(**a**) parameter L*; (**b**) parameter a*; (**c**) parameter b* during accelerated stability. The 0%, 0.5%, 1%, and 2% values mean the different percentages of extract used for the gel formulations. DARK: samples stored in the dark; LIGHT: samples stored in the light. The dotted line marks the boundary between the control group and the samples during long-term stability test. The error bars represent the standard deviation of the triplicates of the analytical measurements.

**Figure 10 ijms-25-03723-f010:**
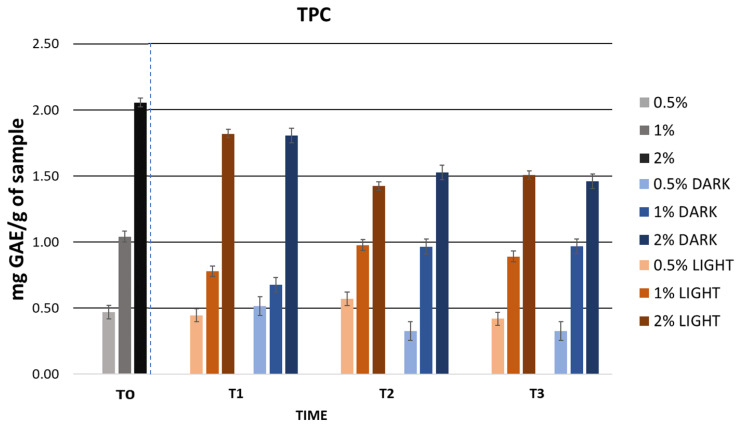
TPC values during long-term stability test. The 0.5%, 1%, and 2% values mean the different percentages of extract used for the gel formulations. DARK: samples stored in the dark; LIGHT: samples stored in the light. The dotted line marks the boundary between the control group and the samples during long-term stability test. The error bars represent the standard deviation of the triplicates of the analytical measurements.

**Table 2 ijms-25-03723-t002:** Evaluation of total phenolic content, antioxidant capacity, free radical scavenging, and enzymatic inhibition of cherry stem extract.

Methodology	CS Extract	EPI	GA	KA	ELA	PHE
TPC (mg GAE/g DE)	131.4 ± 0.4	-	-	-	-	-
FRAP (mmol FeSO_4_/g DE)	0.96 ± 0.01	-	-	-	-	-
TEAC (μmol TE/g DE)	1005.5 ± 0.2	-	-	-	-	-
ORAC (mmol TE/g DE)	2347 ± 0	-	-	-	-	-
O_2_^·−^ (mg/L) ^1^	198 ± 6	70 ± 5	50 ± 3	-	-	-
NO^·^ (mg/L) ^1^	2.3 ± 0.4	0.87 ± 0.02	1.4 ± 0.3	-	-	-
HOCl (mg/L) ^1^	5.4 ± 0.5	0.18 ± 0.01	3.8 ± 0.3	-	-	-
Tyrosinase	690 ± 30 ^2^	-	-	49 ± 6 ^3^	-	-
Elastase	537 ± 16 ^1^	-	-	-	53 ± 5 ^4^	-
HYALase (mg/L)	7.39 ± 0.06 ^1^	167 ± 6 ^5^	102 ± 4 ^5^	-	-	-
Collagenase (mg/L) ^1^	112 ± 8	-	-	-	-	83 ± 2
XOD (mg/L) ^1^	10 ± 1	9 ± 1	-	-	-	-

CS: cherry stem; FRAP: ferric reducing antioxidant power assay; TEAC: Trolox equivalent antioxidant capacity; ORAC: oxygen radical absorbance capacity; GAE: gallic acid equivalent; DE: dry extract; TE: Trolox equivalent. HYALase: hyaluronidase; EPI: epicatechin; GA: gallic acid; KA: Kojic acid; ELA: elastatinal; PHE: 1, 10-phenanthroline. ^1^ Inhibitory concentration at 50% (mg/L). ^2^ Inhibitory concentration at 55% (mg/L). ^3^ % Inhibition at 21.3 mg/L. ^4^ % Inhibition at 51.26 mg/L. ^5^ Inhibitory concentration at 20% (mg/L). Data are means ± standard deviation (n = 3).

**Table 3 ijms-25-03723-t003:** Initial values of the physicochemical parameters of the gels.

Parameters	0%	0.5%	1%	2%
**Viscosity (Pa·s)**				
10 rpm	8.40 ± 0.00	9.00 ± 0.00	11.58 ± 0.03	15.30 ± 0.00
70 rpm	1.67 ± 0.01	1.81 ± 0.00	2.21 ± 0.01	2.76 ± 0.00
**Colour**				
L*	25.73 ± 0.02	24.76 ± 0.04	24.67 ± 0.01	24.65 ± 0.02
a*	−0.58 ± 0.01	−0.46 ± 0.01	−0.08 ± 0.02	−0.08 ± 0.02
b*	0.88 ± 0.06	5.52 ± 0.04	5.64 ± 0.03	6.01 ± 0.02
**pH**	5.92 ± 0.01	5.37 ± 0.01	4.97 ± 0.01	4.66 ± 0.01
**TPC (mg GA/g DE)**	0	0.47 ± 0.01	1.04 ± 0.23	2.05 ± 0.07

Data are means ± standard deviation (n = 3). Rpm: revolutions per minute; GAE: gallic acid equivalent; DE: dry extract; 0%: gel without extract; 0.5%: gel with 0.5% cherry stem extract: 1%: gel with 1% cherry stem extract; 2%: gel with 2% cherry stem extract.

**Table 4 ijms-25-03723-t004:** List of formulation ingredients.

Commercial Name	INCI Name	CAS Code	Function	%
Demineralized water	Water	7732-18-5	Aqueous phase	89.97, 89.47, 88.97, 87.97
Glycerine	Glycerine	56-81-5	Humectant	5
Vegetal extract	-	-	Active ingredient	0, 0.5, 1, 2
Xanthan gum	Xanthan gum	11138-66-2	Emulsifier/Gelling agent	2
Ethanol	Alcohol	64-17-5	Solvent/Preservative	3
Citric acid	Citric acid	77-92-9/5949-29-1	pH regulator	0.03

INCI: International nomenclature of cosmetic ingredients.

## Data Availability

The data presented in this study are available on request from the corresponding author.

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
