# Peer review of "Bioactive Compounds and Potential Health Benefits through Cosmetic Applications of Cherry Stem Extract"

_ijms, 2024, doi:10.3390/ijms25073723_

Round 1

Reviewer 1 Report

Comments and Suggestions for Authors

Comment

Bioactive compounds and potential health benefits through cosmetic applications of cherry stem extract”, by Abigail García-Villegas et al., has been reviewed.

Authors aimed to evaluate cherry stem (CS) extracts as a potential source of phytochemicals of antioxidativeand anti-aging naturally occurred materials also characterize the component of the phytochemicals. In addition, authors assessed the stability of a gel formulation containing CS extracts. This reviewer has questions and comments.

1. Phytochemicals contained in CS have been already characterized. Antioxidant activities of CS is also known. Authors need to demonstrate the novelty of their study.

2. Related Comment 1: Babota M et al, Front Pharmacol 2021, 12:64797 reported Chemical Composition and biological activity of CS extracts. Authors should distinguish Babota M study from authorsstudy.

3. Is the TPC of CS extracts alone (not in a gel formulation) substantially decreased during storage, i.e., to show advantage to use a gel formulation? 

4. Identified phytochemicals are listed in Table 1. Which chemical are major compounds in CS extracts? 

5. Table 2, Unit of tyrosinase and elastase should be reported.

6. Fig. 9: Authors interpret “TPC values are slightly decreased during the first month of storage, however, from the second month onwards the values remained constant…..

Despite this slight decrease, the formulations continued to show interesting TPC values until the end of the study.” But TPC values of 0.5 and 2% formulation are further decreases at later time points.

7. Figs 2-9: Standard deviation is not included. Were assays conducted using one sample? 

8. Table 1 and 2: Data are shown +/-, Are numbers of variation SD or SE? N values should be reported.

Author Response

Authors aimed to evaluate cherry stem (CS) extracts as a potential source of phytochemicals of antioxidative and anti-aging naturally occurred materials also characterize the component of the phytochemicals. In addition, authors assessed the stability of a gel formulation containing CS extracts. This reviewer has questions and comments.

Phytochemicals contained in CS have been already characterized. Antioxidant activities of CS is also known. Authors need to demonstrate the novelty of their study. Related Comment 1: Babota M et al, Front Pharmacol 2021, 12:64797 reported Chemical Composition and biological activity of CS extracts. Authors should distinguish Babota M study from authors‘study.

Thank you very much for your constructive suggestion. The present study is mainly focused on the benefits of cherry stem extract on the skin due to its high antioxidant and anti-enzymatic potential. The novelty relies on the fact that cherry stem extract has never been incorporated previously into a cosmetic gel to evaluate its stability. The approach and the development of a cosmetic formulation incorporating the extract makes the present study different and novel. All this is highlighted in the introduction (Lines 104-108).

In addition, the characterization carried out in this study has been more exhaustive, allowing the detection of more compounds than previously reported. In fact, 62 compounds were annotated for first time for cherry steam extract. This study has shown that the cherry stem sample used has a higher antioxidant potential than other cherry stem samples previously used.

Thank you very much for letting us know about the article by Babota et al. We have considered this reference and have discussed the main differences between our results and those reported in that study. We leave here the main differences between both studies:

The TPC values were significantly lower in that study than those obtained in the present study. Babota et al. obtained TPC values of 37.63 ± 2.75 mg GAE/g extract (CSE) and 19.11 ± 1.52 mg GAE/g extract (CSD) for the cherry stem samples while in the present study a TPC value of 233.2 ± 0.7 mg GAE/g dry extract was obtained. This discussion has been added in the manuscript (Lines 191-194).

Regarding the characterisation of compounds, Babota et al. noted only 27 compounds in the cherry stem extract while 149 compounds were noted in the present article. In addition, 62 compounds were characterised for the first time in the cherry stem extract. This discussion has been added in the manuscript (Lines 153-158).

The result obtained by Babota et al. for tyrosinase enzyme inhibition was much lower than that obtained in the present study. The IC50 values of Babota et al. for the cherry stem were 8.66 mg/ml (CSE) and 3.03 mg/ml (CSD) while the IC55 value for the cherry stem of the present study was 0.69 mg/ml. This discussion has been added to the manuscript (Lines 305-307).

  1. Is the TPC of CS extracts alone (not in a gel formulation) substantially decreased during storage, i.e., to show advantage to use a gel formulation?

No single storage of the extract was carried out. However, if the extract is subjected to high temperatures, the bioactive compounds present in the extract may degrade. The development of a gel is one way of transporting the extract for application to the skin. In addition, the gel could also provide some stability and protection of the extract against extreme conditions during stability testing.

  1. Identified phytochemicals are listed in Table 1. Which chemical are major compounds in CS extracts?

Thank you very much for your question, thanks to it we have improved the content of Table 1. For a better understanding of the most important compounds in the cherry stem extract, a column containing the relative abundances of each of the listed compounds has been added to Table 1. In summary, flavonoids were the most relevant group in the cherry stem extract, mainly the sub-class flavan-3-ols, flavones and proanthocyanidins. This has been discussed in the manuscript (lines 118-122, 727-728, Table 1)

  1. Table 2, Unit of tyrosinase and elastase should be reported.

Thank you very for this comment. Here is the clarification of this point. Tyrosinase and elastase units are indicated in the table legend because the meaning is different depending on whether the extract or control (KA and ELA) are used. These differences are based on the assay’s protocols.

The result for the enzyme tyrosinase is expressed in mg/L for the extract. As indicated in the table and in the legend, it took 690 mg/L of cherry stem extract to inhibit 55% of the enzyme. However, the 49 % value expressed for kojic acid (KA) is the percentage of inhibition obtained for a KA concentration of 21.3 mg/L.

Similarly, it occurs for the elastase enzyme. (mg/l for the extract vs % inhibition for ELA at 51.26 mg/L). Therefore, the tyrosinase and elastase units are reported that way in the legend rather than in the main table.

  1. Fig. 9: Authors interpret “TPC values are slightly decreased during the first month of storage, however, from the second month onwards the values remained constant…..

Despite this slight decrease, the formulations continued to show interesting TPC values until the end of the study.” But TPC values of 0.5 and 2% formulation are further decreases at later time points.

Thank you very much for your suggestions and corrections. Indeed, Figure 9 shows how the TPC values decreased slightly during the first two months of storage. From the second month on, the values remained almost constant with the exception of the 0.5% formulation stored in light where the TPC value continues to decrease until T3. These issues were corrected in the paper (Lines 503-505).

  1. Figs 2-9: Standard deviation is not included. Were assays conducted using one sample?

We appreciate your comment. Figures 2-9 show the results of the stability tests. In this preliminary study, replicates of each of the conditions were not performed due to resource limitations of the study. Due to the project limitations, it was preferred to evaluate different conditions of the formulations with the modification of the % of extract added (0.5, 1 and 2%) instead of only one condition of % of extract with replicates. In any case, technical replicates were carried out for each condition for each of the measurements carried out, showing consistency of the analytical process used (low RSD). As a future perspective, we would like to validate these results also using extracts grown in other geoclimatic conditions.

  1. Table 1 and 2: Data are shown +/-, Are numbers of variation SD or SE? N values should be reported.

In the paper +/- indicates standard deviation with n=3. It has been clarified in the manuscript.

Reviewer 2 Report

Comments and Suggestions for Authors

Dear authors,

The manuscript "Bioactive compounds and potential health benefits through cosmetic applications of cherry stem extract", entails the use of a cherry stem extract rich in bioactive compounds for cosmetic applications and the prove of concept by incorporating the extract in a formulation and evaluating its stability.

The manuscript is well organized and in general it is well written.

In all the manuscript there are no photos of the extract or the hydrogel formulations obtained by the incorporation of the extract. Since there is a mention the extract color it would be useful to see the aspect and the impact it has in the formulation. Color most of the times is an important parameter in formulations acceptability by the consumers' and specially the industry.

The bioactivities of the extract were well characterized however no assay was performed in order to assess the safety of the extract. Namely citcompatbility and sensitization at effective concentration.

Regarding the concentrations of the extract tested in different bioactivities in page 13 there is the use of units that are not clear " 50 % (mg/L)". % is g/100mL and it is not a correct to use % mg/L. And all the units should be uniformed as % or mg/L.

Regarding the stability under accelerated conditions it is not clear the references used to select that protocol. In fact, there is ISO/TR18811 Cosmetics — Guidelines on the stability testing of cosmetic products (2018) that regulates the formulation stability protocol under standard and accelerated conditions and is used in cosmetic industry. 

Regarding the stability of the formulations microbial quality of the formulations should be assessed as other parameter. Since using natural extracts the risk of contamination is higher and the microbial quality of the formulation through time should be analyzed. Moreover, since the formulation contains etanol 3% as preservative. This is not an adequate preservative for cosmetic formulations and is not used in any cosmetic formulation as preservative, specially at 3%. If the formulations are contaminated during the stability assay it will affect the parameters analyzed, eg. pH, color....

In Page 16, line 341 authors talk about the pseudo plastic behavior of cosmetic formulations. However, authors only evaluated the dynamic viscosity a "IKA ROTAVISC me-vi". This does not allow to infer about the pseudo plastic behavior of the material. This analysis is performed in a shear stress vs shear rate analysis.

In page 21, line 465, there is the mention of the times tested but not the information of what period of time corresponds to T1, T2 and T3.

In page 28, line 663, table 1 (please check table numeration, this should be table 4) glycerin is considered an emollient, however, glycerin is not emollient it is an humectant. Emollients are oils.

In page 18, line 414 there is the mention of phase separation concept. Since the formulation used is an hydrogel formulation, so it is a formulation where there is not 2 phases like an emulsion. There is a reticulate matrix of an hydrogel and therefore you cannot talk about phase separation. You can analyse the stability of the reticulated matrix and the loss of integrity the gel...

Comments on the Quality of English Language

Check the spelling and the number of the tables.

Author Response

Dear authors,

The manuscript "Bioactive compounds and potential health benefits through cosmetic applications of cherry stem extract", entails the use of a cherry stem extract rich in bioactive compounds for cosmetic applications and the prove of concept by incorporating the extract in a formulation and evaluating its stability.

The manuscript is well organized and in general it is well written.

Thank you very much for the review and your suggestions.

In all the manuscript there are no photos of the extract, or the hydrogel formulations obtained by the incorporation of the extract. Since there is a mention the extract color it would be useful to see the aspect and the impact it has in the formulation. Color most of the times is an important parameter in formulations acceptability by the consumers' and specially the industry.

During the study, we did not have the means or the necessary photographic equipment to take high quality pictures where the small colour variations in the gels can be seen. The photographs available to us are of poor quality and are not suitable for publication in this article. However, it is important to note that the variations in the colour parameters L*, a* and b* are minimal and cannot be seen with the naked eye. For future studies, consideration will be given to the provision of high-quality images through which the characteristics of the formulations can be perceived.

The bioactivities of the extract were well characterized however no assay was performed in order to assess the safety of the extract. Namely citcompatbility and sensitization at effective concentration.

Many thanks for your suggestions. This is a preliminary study to see if the extract, in addition to being bioactive, is stable in a cosmetic formulation. Cytocompatibility and sensitisation studies will be carried out in the future to better understand the extract on the skin and its possible uses in the cosmetic industry.

Regarding the concentrations of the extract tested in different bioactivities in page 13 there is the use of units that are not clear " 50 % (mg/L)". % is g/100mL and it is not a correct to use % mg/L. And all the units should be uniformed as % or mg/L.

It is the concentration expressed in mg/L necessary to inhibit 50% of the enzymatic activity.

Regarding the stability under accelerated conditions, it is not clear the references used to select that protocol. In fact, there is ISO/TR18811 Cosmetics — Guidelines on the stability testing of cosmetic products (2018) that regulates the formulation stability protocol under standard and accelerated conditions and is used in cosmetic industry.

Thank you very much for this very important appreciation. The stability studies were performed following the guidelines indicated in the ISO/TR18811 “Cosmetics - Guidelines on the stability testing of cosmetic products (2018)”. We have mentioned and clarified this point in the new version of the manuscript. (Lines 697-700)

Regarding the stability of the formulations microbial quality of the formulations should be assessed as other parameter. Since using natural extracts, the risk of contamination is higher and the microbial quality of the formulation through time should be analyzed. Moreover, since the formulation contains etanol 3% as preservative. This is not an adequate preservative for cosmetic formulations and is not used in any cosmetic formulation as preservative, specially at 3%. If the formulations are contaminated during the stability assay it will affect the parameters analyzed, eg. pH, color....

Microbial quality is a crucial factor when developing a cosmetic. The present work aims to develop a formula with cherry stem extract in a preliminary way and to observe its stability. As a future perspective, one of the main objectives for further progress in this field will be to carry out microbiological tests on the cosmetic prototypes to be developed.

On the other hand, ethanol can be found as a cosmetic ingredient in the INCI with CAS number: 64-17-5, where it is indicated that one of its main functions is that of an antimicrobial agent, helping to stop the growth of microorganisms on the skin. Currently, ethanol is not yet regulated in cosmetics. Therefore, the % used in this study is low.

In Page 16, line 341 authors talk about the pseudo plastic behavior of cosmetic formulations. However, authors only evaluated the dynamic viscosity a “IKA ROTAVISC me-vi”. This does not allow to infer about the pseudo plastic behavior of the material. This analysis is performed in a shear stress vs shear rate analysis.

Thank you very much for your comments for the improvement of this article. With regard to pseudoplastic fluids, these are whose viscosity decreases with increasing applied strain rate [1]. In this case, the viscosity was measured at two different speeds, 10 and 70 rpm. The results showed that the viscosity of the gel decreased with increasing speed. The viscosity was higher at 10 rpm and lower at 70 rpm.

  1. Martinez, R.M.; Rosado, C.; Velasco, M.V.R.; Lannes, S.C.S.; Baby, A.R. Main Features and Applications of Organogels in Cosmetics. Int. J. Cosmet. Sci. 2019, 41, 109–117, doi:10.1111/ics.12519.

In page 21, line 465, there is the mention of the times tested but not the information of what period of time corresponds to T1, T2 and T3.

Thank you for this appreciation. The times T0, T1, T2 and T3 were indicated in more detail in the paper (lines 474-477).

In page 28, line 663, table 1 (please check table numeration, this should be table 4) glycerin is considered an emollient, however, glycerin is not emollient it is an humectant. Emollients are oils.

It has been corrected in the paper. Thank you very much.

In page 18, line 414 there is the mention of phase separation concept. Since the formulation used is an hydrogel formulation, so it is a formulation where there is not 2 phases like an emulsion. There is a reticulate matrix of an hydrogel and therefore you cannot talk about phase separation. You can analyse the stability of the reticulated matrix and the loss of integrity the gel...

It has been corrected in the paper. Thank you very much.

Round 2

Reviewer 1 Report

Comments and Suggestions for Authors

The authors revised manuscript in response to reviewers’ comments. Manuscript has been improved. This reviewer still have a few comments.

Comment 1, related prior comment 3 “Is the TPC of CS extracts alone (not in a gel formulation) substantially decreased during storage, i.e., to show advantage to use a gel formulation?”

Response: No single storage of the extract was carried out. However, if the extract is subjected to high temperatures, the bioactive compounds present in the extract may degrade. The development of a gel is one way of transporting the extract for application to the skin. In addition, the gel could also provide some stability and protection of the extract against extreme conditions during stability testing.

Auhtors should conduct additional study to check instability of CS extracts solution to demonstrate importance of authors’ study. Without this additonal result, it is difficult to show significance of authors’ study.

Comment 1, related prior comment 7 “Figs 2-9: Standard deviation is not included. Were assays conducted using one sample? 

Response: We appreciate your comment. Figures 2-9 show the results of the stability tests. In this preliminary study, replicates of each of the conditions were not performed due to resource limitations of the study. Due to the project limitations, it was preferred to evaluate different conditions of the formulations with the modification of the % of extract added (0.5, 1 and 2%) instead of only one condition of % of extract with replicates. In any case, technical replicates were carried out for each condition for each of the measurements carried out, showing consistency of the analytical process used (low RSD). As a future perspective, we would like to validate these results also using extracts grown in other geoclimatic conditions.

Data of technical replicattion can be useful, i.e., report Mean+/-SD. 

Author Response

Comment 1, related prior comment 3 “Is the TPC of CS extracts alone (not in a gel formulation) substantially decreased during storage, i.e., to show advantage to use a gel formulation?”

Response: No single storage of the extract was carried out. However, if the extract is subjected to high temperatures, the bioactive compounds present in the extract may degrade. The development of a gel is one way of transporting the extract for application to the skin. In addition, the gel could also provide some stability and protection of the extract against extreme conditions during stability testing.

Authors should conduct additional study to check instability of CS extracts solution to demonstrate importance of authors’ study. Without this additonal result, it is difficult to show significance of authors’ study.

We greatly appreciate your suggestions for this study. Thanks to them, we have been able to delve deeper into the stability of cherry stem extract.

We conducted an additional study where the cherry stem extract alone, diluted in an 80:20 ethanol and water solution (not incorporated into the cosmetic formulation), was subjected to various conditions (centrifugation, temperature ramp, and heating-cooling cycles) to evaluate its stability. After storage, it was observed that the TPC values of the extract decreased in all cases, especially after the temperature ramp and centrifugation (see Figure 2). Therefore, we can conclude that the stability of the TPC value of cherry stem extract is significantly affected by temperature.

However, it was noted that the TPC values of the developed cosmetic formulations containing the extract hardly varied. Therefore, we can infer that the extract could be protected by the formulation, and its stability could be favoured by it.

Comment 1: related prior comment 7 “Figs 2-9: Standard deviation is not included. Were assays conducted using one sample?

Response: We appreciate your comment. Figures 2-9 show the results of the stability tests. In this preliminary study, replicates of each of the conditions were not performed due to resource limitations of the study. Due to the project limitations, it was preferred to evaluate different conditions of the formulations with the modification of the % of extract added (0.5, 1 and 2%) instead of only one condition of % of extract with replicates. In any case, technical replicates were carried out for each condition for each of the measurements carried out, showing consistency of the analytical process used (low RSD). As a future perspective, we would like to validate these results also using extracts grown in other geoclimatic conditions.

Thank you very much for your feedback to improve this paper.

The standard deviation of the triplicates was added in all figures of the article.

Reviewer 2 Report

Comments and Suggestions for Authors

I am happy with the alterations made by the authors.

Author Response

Thank you very much for your comment.

Round 3

Reviewer 1 Report

Comments and Suggestions for Authors

This reviewer has not further comment.

Author Response

Thank you very much.